# Biomimetic mercury immobilization by selenium functionalized polyphenylene sulfide fabric

Hailong Li [1], Fanyue Meng[1], Penglin Zhu[1], Hongxiao Zu[1], Zequn Yang[1], Wenqi Qu[1] & Jianping Yang [1] ✉

Highly efficient decontamination of elemental mercury ($Hg^0$) remains an enormous challenge for public health and ecosystem protection. The artificial conversion of $Hg^0$ into mercury chalcogenides could achieve $Hg^0$ detoxification and close the global mercury cycle. Herein, taking inspiration from the bio-detoxification of mercury, in which selenium preferentially converts mercury from sulfoproteins to HgSe, we propose a biomimetic approach to enhance the conversion of $Hg^0$ into mercury chalcogenides. In this proof-of-concept design, we use sulfur-rich polyphenylene sulfide (PPS) as the $Hg^0$ transporter. The relatively stable, sulfur-linked aromatic rings result in weak adsorption of $Hg^0$ on the PPS rather than the formation of metastable HgS. The weakly adsorbed mercury subsequently migrates to the adjacent selenium sites for permanent immobilization. The sulfur-selenium pair affords an unprecedented $Hg^0$ adsorption capacity and uptake rate of 1621.9 mg g$^{-1}$ and 1005.6 µg g$^{-1}$ min$^{-1}$, respectively, which are the highest recorded values among various benchmark materials. This work presents an intriguing concept for preparing $Hg^0$ adsorbents and could pave the way for the biomimetic remediation of diverse pollutants.

Mercury exposure causes many serious health issues in humans and is known as Minamata disease[1,2]. Even though a global agreement, i.e., the Minamata Convention, was recently forged to reduce the threat of mercury to humans[3], total global anthropogenic mercury emissions are still higher than 2100 tons per year[4,5]. The mercury into the environment is primarily discharged from industrial flue gases. Among various mercury species, elemental mercury ($Hg^0$) is the most difficult to remove due to its high volatility and water insolubility[6]. The interconvertible $Hg^0$ and methyl mercury create great hazards to human health[7]. Thus, it is imperative to develop advanced technologies for $Hg^0$ removal from industrial flue gases.

Since ultrastable mercury chalcogenides are generally natural carriers of mercury and original sources accounting for anthropogenic $Hg^0$ emission[8,9], conversion of $Hg^0$ back to its chalcogenide forms is regarded as a golden reaction for detoxification of $Hg^0$ pollution and closing of the global mercury cycle[10–13]. To realize this conversion

process, the use of mineral chalcogenides seems to be a straightforward pathway that mimics the geological deposition of mercury in nature, a technique that has been developed for several years and extensively explored since 2016[5,14–17]. Both sulfur-laden and selenium-laden sorbents have been studied, and most selenium-laden sorbents outperformed the sulfur-laden sorbents because the binding constant ($K_a$) of selenium and mercury is $10^{45}$, approximately $10^6$ times higher than that of mercury and sulfur[12,18,19]. This high affinity constant made the $Hg^0$ adsorption capacities of selenium-based adsorbents are at least 100 times higher than those of sulfur adsorbents with similar chelating site coverage ratios[11,14,17,20]. However, the adsorption of $Hg^0$ on selenium adsorbents still does not reach 40% of the theoretical values[12]. There is still the possibility of increasing the $Hg^0$ adsorption capacities of chalcogenide-based adsorbents far beyond this limit with a 1:1 stoichiometric Se/Hg ratio. To achieve this goal, we believe that it is necessary to take a fact that has long been overlooked into

[1]School of Energy Science and Engineering, Central South University, Changsha 410083, China. ✉e-mail: jpyang@csu.edu.cn

consideration, i.e., detoxification of mercury in living organisms with the help of chalcogenides has a much higher efficiency than that in the ambient environment. Thus, if we can learn from mercury detoxification in the human body through rational material design, it may be a feasible way to develop efficient adsorbents for $Hg^0$ removal from anthropogenic sources.

It is worth noting that the toxicity of mercury for organisms is due to covalent binding to sulfhydryl goups (-SH) in enzymes[21]. Selenium is the optimal detoxicant for mercury poisoning[22]. Seleno-mercury antagonism results because selenium preferentially converts mercury from -SH complexes to HgSe owing to the strong Hg-Se binding affinity. Meanwhile, HgSe has a solubility constant of $1.0 \times 10^{-59}$, and it is eliminated from the body through metabolism to recover the vitality of the mercury-destroyed enzymes[23]. This biological antagonism implies that mercury might migrate from sulfur when encountering selenium ligands. Inspired by this law, we proposed a feasible way to enhance the accessibility of selenium, i.e., $Hg^0$ is provisionally adsorbed on a substance with abundant sulfur sites, and then the weakly adsorbed mercury migrates from the sulfur sites to adjacent selenium sites for permanent immobilization (as illustrated in Fig. 1). To confirm this assumption, and after screening various sulfur-containing supports, PPS fabrics consisting of aromatic rings linked with sulfur were selected. The molecular structure of PPS provides plentiful sulfur sites on the fabrics[24–26]. Coincidentally, $Hg^0$ is weakly adsorbed by the sulfurs in PPS but does not undergo firm immobilization to form HgS, since the relatively stable organosulfur groups in PPS require adequate activation energy at a high temperature to induce bonding with Hg. This implies that the sulfur in PPS primarily acts as a buffer in transporting $Hg^0$, while the tangled structure connected by many fibers provides continuous transport channels for mercury migration to selenium ligands. Thus, selenium ligands with high activity and sufficient accessibility can be constructed simultaneously to provide an ideal $Hg^0$ adsorbent.

Herein, a scalable in situ synthetic method was developed to prepare selenium-functionalized PPS (Se/PPS-I) for $Hg^0$ removal from industrial flue gases. Compared to the common postsynthetic methods (i.e., first synthesizing selenium powder and then coating it on a support), the in situ synthetic method provided a high density of available selenium ligands for mercury transport from the inherent sulfur sites on PPS, thus increasing the $Hg^0$ adsorption capacity of Se/PPS-I. The resultant Se/PPS-I exhibited a mercury adsorption capacity and uptake rate of $1621.9\ mg\ g^{-1}$ and $1005.6\ \mu g\ g^{-1}\ min^{-1}$, respectively. Thus, the adsorbent properties were rationally engineered, which demonstrates the feasibility of converting $Hg^0$ into mercury chalcogenides based on a biomimetic pathway.

## Results

### Rational design and synthesis of Se/PPS

PPS displays a tangled structure (Supplementary Fig. 1). Figure 2a–c and Supplementary Fig. 2a show that a small amount of selenium was randomly distributed on PPS when prepared by a postsynthetic method (denoted Se/PPS-P). In contrast, when using the same amount of selenium precursor, the in situ synthetic method generated a dense and homogeneous coverage of selenium on Se/PPS-I, as displayed in Fig. 2d–f and Supplementary Fig. 2b. This design easily fabricated Se/PPS-I fabrics measuring 100 cm ×200 cm, thus implying great potential for scaled processing (Supplementary Fig. 3). Quantum chemical simulations were conducted to reveal the post- and in situ synthetic processes. The differences between the post- and in situ synthetic processes was the adsorbate (i.e., Se and $SeO_3^{2-}$) on the PPS. The structures of the PPS monomer, Se molecule, and $SeO_3^{2-}$ anion are shown in Supplementary Fig. 4. Electrostatic interactions, which are

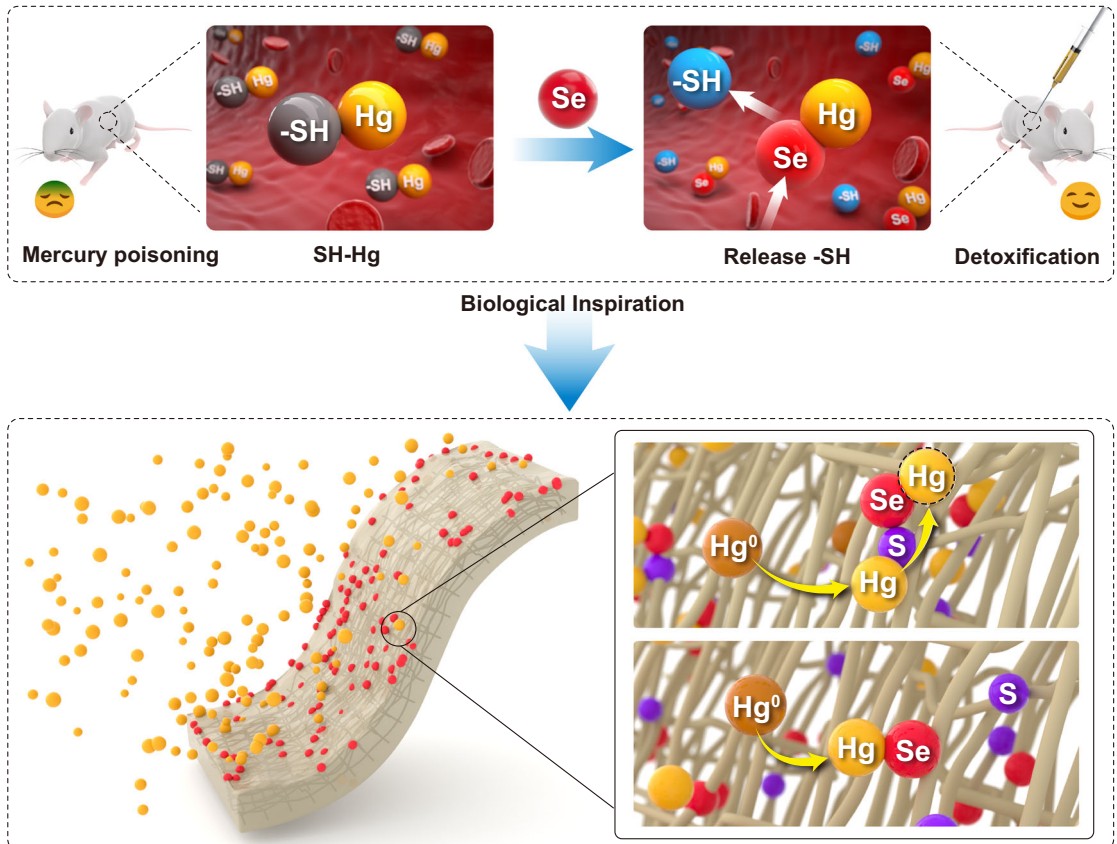

**Fig. 1 | Biomimetic mindset for increasing $Hg^0$ immobilization on the selenium adsorbent.**

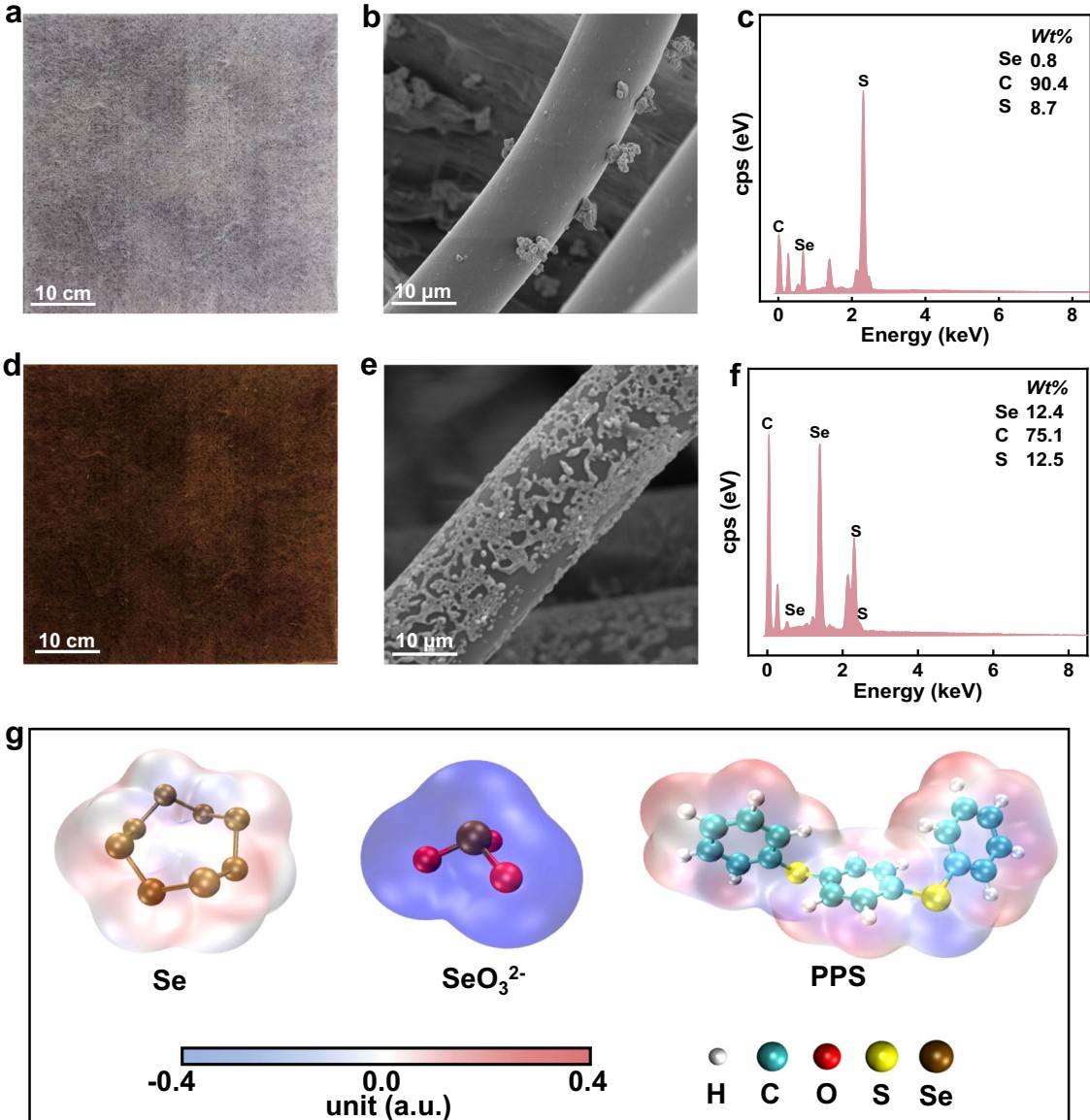

**Fig. 2 | Selenium surface coverage on Se/PPS prepared with different methods and the differences between post- and in situ synthetic processes. a** Photograph, **b** SEM image and **c** EDS maps of Se/PPS-P, **d** Photograph, **e** SEM image and **f** EDS maps of Se/PPS-I and **g** ESP distributions of Se, $SeO_3^{2-}$, and PPS.

determined by the electrostatic surface potential (ESP) distribution, usually play decisive roles in combining molecules in solution. As shown in Fig. 2g, the distribution of ESP in the Se molecule was weak and uniform due to limited charge transfer between the Se atoms. In contrast, the negative charge of the $SeO_3^{2-}$ ion and the ESP distribution generated nucleophilicity. Thus, compared with the Se molecule, $SeO_3^{2-}$ ions are more inclined to approach a system with a positive ESP distribution. Coincidentally, the ESP distribution of the PPS monomer suggested the electrophilicity was due to charge transfer from H atoms to C and S atoms. Thus, compared with Se, $SeO_3^{2-}$ was more easily anchored by the H atoms of PPS via Coulombic forces, and $SeO_3^{2-}$ can be reduced to elemental selenium with a reductant (i.e., glutathione, GSH) under alkaline conditions. Meanwhile, PPS displayed super-hydrophobicity (Supplementary Fig. 5), thus leading to preferential adsorption of the $SeO_3^{2-}$ from solution and ensuring a highly densely populated selenium precursor. The preferential adsorption of $SeO_3^{2-}$ accelerated the solid–liquid interfacial assembly with PPS, thus allowing targeted anchoring of selenium for a more uniform surface coverage and higher adhesiveness of the PPS.

## Characterizations of Se/PPS

The selenium powder showed an aggregated morphology (shown in Supplementary Fig. 6), which might hinder the diffusion of mercury and lead to insufficient accessibility of the selenium[27]. The morphology of Se/PPS-I was easily controlled by adjusting the in situ synthetic conditions (i.e., the NaOH and $SeO_3^{2-}$ concentrations). Figure 3a shows that the PPS surface remained smooth after adsorbing $SeO_3^{2-}$ (i.e., without adding NaOH), while the distribution of selenium in the energy disperse spectroscopy (EDS) image demonstrated successful anchoring of $SeO_3^{2-}$ onto PPS (shown in Supplementary Fig. 7). Figure 3b-e shows that with increasing NaOH concentrations from 0% to 1%, the growth of selenium was accelerated and various morphologies were created, including particles, domes, and films. This suggested that the reduction of $SeO_3^{2-}$ to elemental selenium resulted from the alkalinity[28]. Supplementary Fig. 8a shows the selenium content on Se/PPS-I measured by inductively coupled plasma mass spectrometry (ICP–MS). The selenium loading on Se/PPS-I displayed a non-monotonic increasing tendency with increasing NaOH dosage, which was ascribed to the different selenium nucleation kinetics with

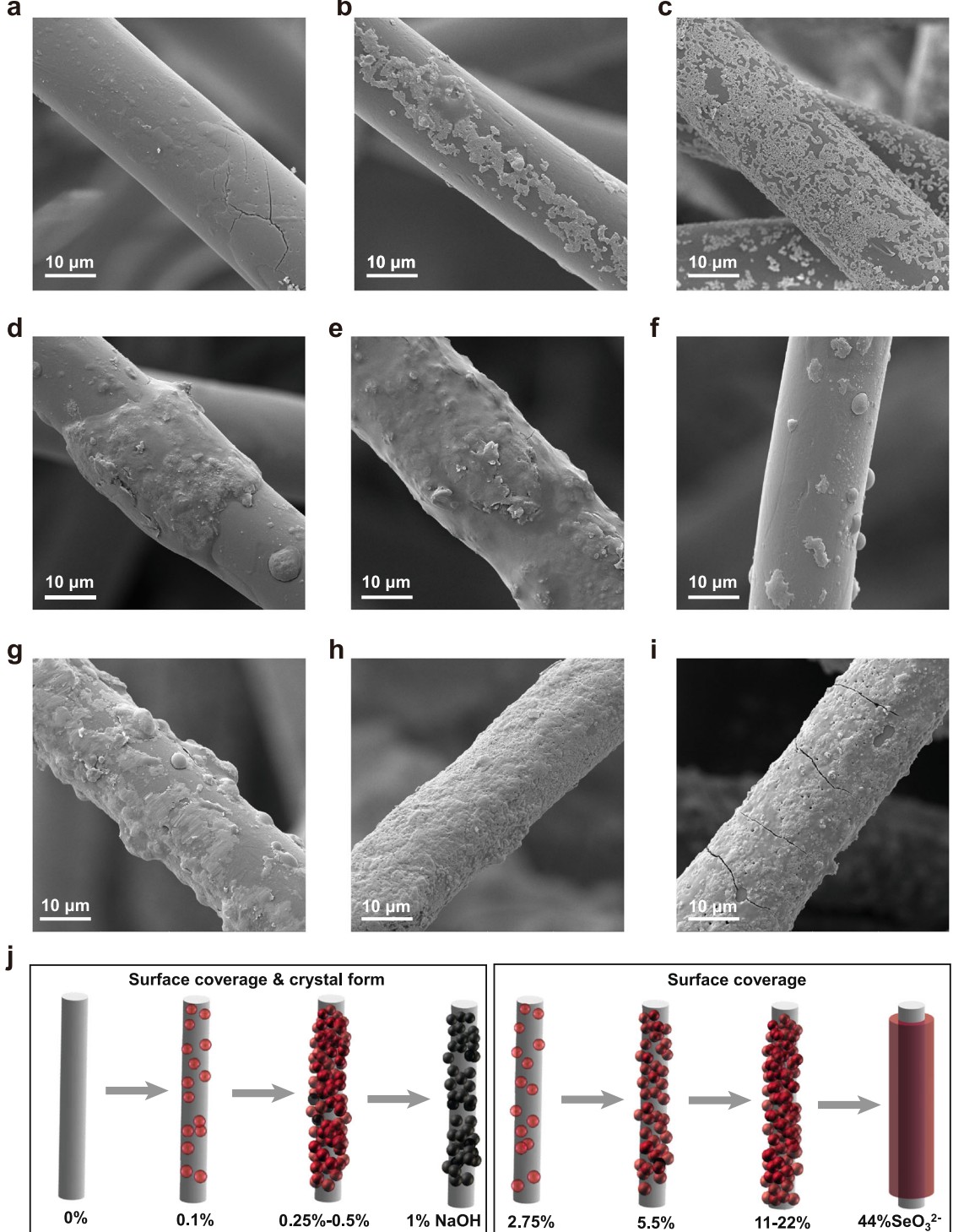

**Fig. 3 | Morphologies of Se/PPS-I samples prepared under different conditions.** SEM images of Se/PPS-I prepared with different NaOH concentrations (SeO$_3^{2-}$ concentration of 5.5%) (**a**) 0%, (**b**) 0.1%, (**c**) 0.25%, (**d**) 0.5%, (**e**) 1%, and different SeO$_3^{2-}$ concentrations (NaOH concentration of 0.25%) (**f**) 2.75%, (**g**) 11%, (**h**) 22%, (**i**) 44%, (**j**) Schematic illustration of the effects of NaOH and SeO$_3^{2-}$ concentration in preparing Se/PPS-I.

different alkalinities[29]. Crystals are generally formed in two stages, i.e., nucleation and growth. The elemental selenium seeds cannot firmly grow on PPS if the nucleation kinetics are too fast. Thus, some of the selenium was exfoliated from PPS and entered the solution during the shocking process. In addition, the NaOH concentration affected the crystalline phase of selenium on Se/PPS-I. As displayed in Supplementary Fig. 9, the selenium on Se/PPS-I was converted from red (monoclinic crystalline form) to black (hexagonal crystalline form) under high alkalinity, which might display different Hg$^0$ adsorption capacities.

The SeO$_3^{2-}$ concentration directly determined the selenium precursor formed on PPS to produce different selenium surface coverages, which was substantiated by the selenium contents measured via ICP–MS (Supplementary Fig. 8b). Only sparse and patchy selenium particles appeared on the PPS with a low SeO$_3^{2-}$ concentration of 2.75% (shown in Fig. 3f), which was most likely due to insufficient precursor

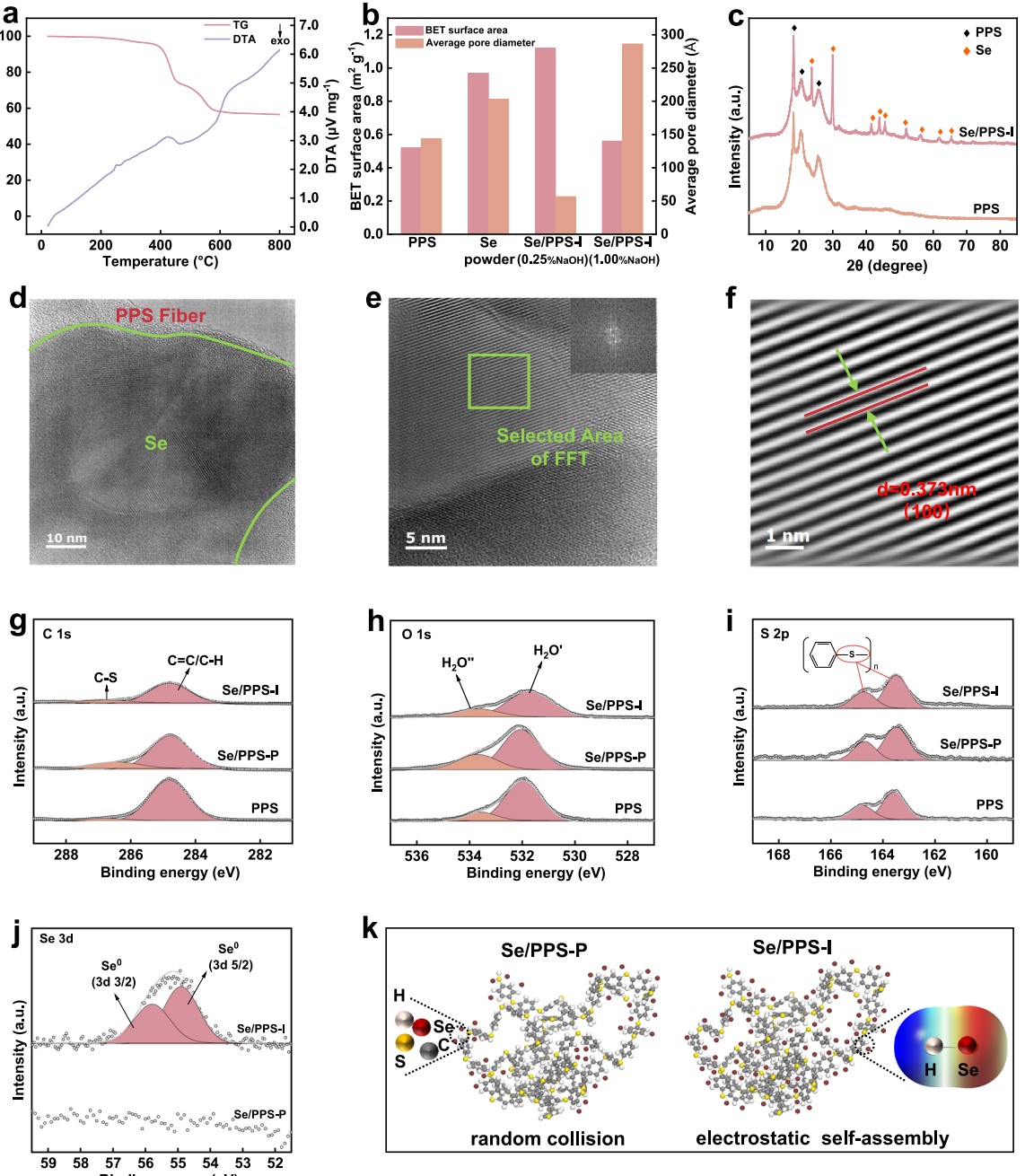

**Fig. 4 | Thermal stability, pore structure, crystallinity, surface chemistry and assembly mechanism of Se/PPS-P and Se/PPS-I. a** TG-DTA curve of Se/PPS-I, **b** BET surface area and average pore size, **c** XRD pattern, **d**, **e** HRTEM, **f** IFFT image, **g** C 1s, **h** O 1s, **i** S 2p and **j** Se 3d XPS spectra and **k** schematic illustration of the assembly mechanism for Se/PPS.

for selenium nucleation and continued growth. With an increase in the $SeO_3^{2-}$ concentration to 5.5%, abundant selenium particles were uniformly dispersed on the PPS (shown in Fig. 3c). Fresher nuclei formed on the primary selenium particles, thus leading to the formation of domes or dense films with further increases in the $SeO_3^{2-}$ concentration to 11%, 22%, and 44% (shown in Fig. 3g–i). A coupled "capturing" mechanism for selenium in situ growth on PPS could explain this phenomenon, as illustrated in Fig. 3j. Appropriate alkalinity was required to reduce $SeO_3^{2-}$ to elemental selenium, while an appropriate reduction rate was needed to generate an adequate amount and the desired crystalline form of selenium. At low $SeO_3^{2-}$ concentrations, a scattered coating was formed on the PPS since the tangled structure of PPS formed a dispersed $SeO_3^{2-}$ system, which did not facilitate the

formation of a supersaturated ion interface layer[30]. In contrast, with a high $SeO_3^{2-}$ concentration, more selenium seeds were generated and aggregated to form a dense selenium coating. Thus, abundant active ligands with the proper crystalline form and surface distribution were produced by adjusting the synthetic conditions to provide an ideal $Hg^0$ adsorbent.

Figure 4a and Supplementary Fig. 10 show the thermogravimetric coupled with differential thermal analysis (TG-DTA) curves for pristine PPS, selenium powder, and Se/PPS-I. As shown, there were three stages of weight loss for Se/PPS-I. The first step (~0.74%) occurred below 100 °C and corresponded to the loss of physically adsorbed water and free guest water. The second step (~3.84%) between 200 and 380 °C was attributed to the volatilization of elemental selenium, while the

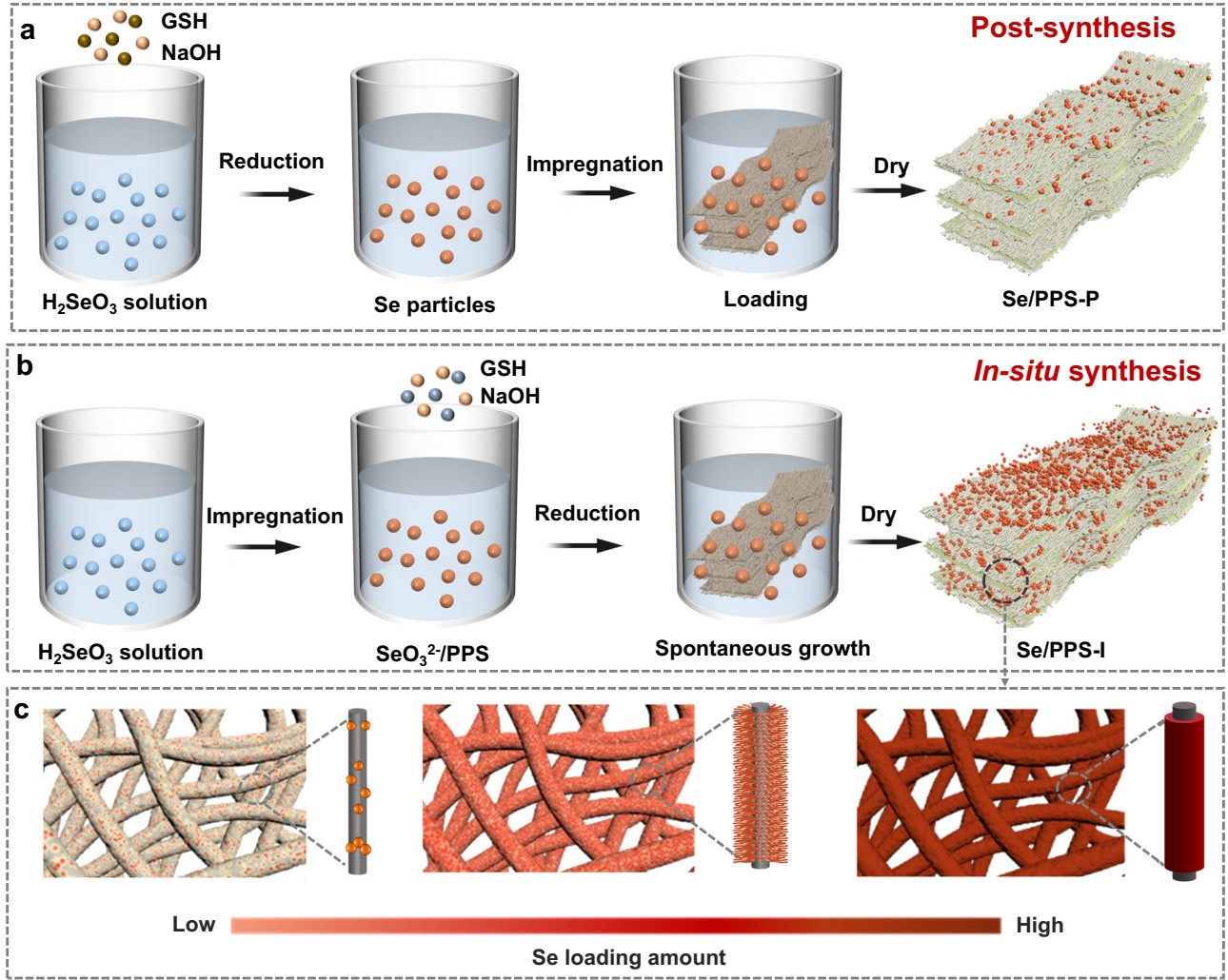

**Fig. 5 | Diagrammatical illustration of the formation of Se/PPS. a** The postsynthetic route to Se/PPS, **b** the in situ synthetic route to Se/PPS, and **c** the surface coverage of selenium with different loading amounts.

weight loss above 380 °C originated from the decomposition of organic matter in the PPS[31,32]. Meanwhile, the stability of selenium on Se/PPS during long-term application was tested. As shown in Supplementary Fig. 11a, the weight change of Se/PPS after purging with $N_2$ for 14 days could be negligible. The amounts of selenium on Se/PPS before and after purging for 14 days were 2.83% and 2.81% (all samples were determined three times), respectively, demonstrating that selenium was firmly anchored on the PPS (as illustrated in Supplementary Fig. 11b). Therefore, it is reasonable to believe that Se/PPS-I can be used as a filter bag in a bag-type dust collector to achieve simultaneous removal of $Hg^0$ and dust from flue gases.

The Brunauer–Emmett–Teller (BET) surface area and average pore diameter of Se/PPS-I are shown in Fig. 4b. As shown, compared with those of PPS and selenium powder, the surface area of Se/PPS-I (with a NaOH concentration of 0.25%) was higher. Meanwhile, Se/PPS-I showed a lower average pore diameter. Compared with the macropores in the PPS and selenium powder, the mesopores of Se/PPS-I favored $Hg^0$ diffusion[33]. The NaOH concentration also affected the surface area and average pore diameter of Se/PPS-I, in which a higher NaOH concentration created a lower surface area and higher average pore diameter due to the formation of a dense selenium layer.

The crystallinity of Se/PPS-I was determined by X-Ray Diffraction (XRD). As shown in Fig. 4c, only elemental selenium was observed in the XRD pattern, while no other species containing selenium were detected. The structure of Se/PPS-I was also investigated by

transmission electron microscopy (TEM). Figure 4d shows the interface between selenium and the PPS fibers. The EDS maps shown in Supplementary Fig. 12 indicated successful immobilization and uniform dispersal of the selenium layer onto the PPS fibers. Additionally, homogeneous distribution of the sulfur in PPS was also observed. The adjacent distributions of sulfur and selenium provided channels for mercury migration between these two sites. Figure 4e, f shows the high-solution TEM (HRTEM) and inverse Fast Fourier transform (IFFT). The measured layer spacing was 0.373 nm, which matched the lattice fringe for the (100) surface of elemental selenium.

The surface chemistries of PPS, Se/PPS-P, and Se/PPS-I were determined by X-ray photoelectron spectroscopy (XPS). The C 1s spectrum contained C-S and C=C/C-H signals with binding energies of 286.9 and 284.8 eV, respectively (shown in Fig. 4g), which were consistent with the composition of the PPS. Figure 4h reveals that the oxygens in PPS and the Se/PPS samples originated from adsorbed water, which is inevitable during exposure to air or aqueous solutions. The S 2p spectrum indicated the presence of organosulfur species in PPS, and the peaks were located at 163.6 and 164.8 eV (shown in Fig. 4i)[34]. Figure 4j shows that two peaks appeared at approximately 54.9 and 55.8 eV in the Se 3d spectrum of Se/PPS-I, which were assigned to elemental selenium[12,35]. No other valence states of selenium were detected in the Se 3d spectrum, demonstrating that the selenium precursor (i.e., $SeO_3^{2-}$) was completely converted to elemental selenium by in situ chemical reduction. The selenium signal was not

observed in the Se/PPS-P spectrum owing to the low loading amount. According to the ESP distribution of PPS, the elemental selenium placed on PPS during the postsynthetic method was primarily anchored via van der Waals forces, suggesting that electron transfer did not appear in the synthetic process, as illustrated in Fig. 4k. However, electron transfer could have occurred when adsorbing the selenium precursor (i.e., $SeO_3^{2-}$) on PPS during the in situ synthetic method owing to the adverse ESP distributions of $SeO_3^{2-}$ and PPS. The positive ESP on PPS was mainly located around the H atoms, suggesting that electron transfer occurred between the H atoms and $SeO_3^{2-}$ ions. As a result, the valence states for C, O, and S in PPS changed insignificantly after anchoring selenium either via postsynthetic or in situ synthetic methods.

Based on the above characterization data, the formation of Se/PPS-P and Se/PPS-I is illustrated in Fig. 5. The postsynthetic process was started by coating a suspension of selenium particles, and deposition on PPS was realized via random collisions (Fig. 5a). As a result, only sparse and patchy selenium particles appeared on the Se/PPS prepared by the postsynthetic method. In this scenario, even if $Hg^0$ was weakly adsorbed on the PPS, there were insufficient selenium ligands to bind mercury that migrated from the sulfur sites. The weakly adsorbed mercury might be discharged from the PPS into the flue gas. The in situ synthetic method (Fig. 5b) overcame these drawbacks, and $SeO_3^{2-}$ was first anchored on PPS to serve as a selenium precursor, and then was reduced to elemental selenium by a reductant (i.e., GSH) under alkaline conditions. The electrostatically directed assembly of selenium on PPS was realized due to the adverse ESP distribution derived from well-organized adsorption of $SeO_3^{2-}$ onto the PPS. Rational adjustment of the population density and the selenium distribution was achieved by adjusting the $SeO_3^{2-}$ and NaOH concentrations (Fig. 5c), thus providing adequate selenium for binding mercury.

## Robust $Hg^0$ adsorption performance

$Hg^0$ adsorption by PPS, selenium powder, $SeO_3^{2-}$/PPS, Se/PPS-P, and Se/PPS-I were tested to evaluate the effects of the selenium surface coverage and availability. As shown in Fig. 6a, when pristine PPS was employed, the accumulated $Hg^0$ adsorption efficiency was 12.1% within 120 min. The $Hg^0$ capture ability of pristine PPS was attributed to the sulfur in PPS. Although the formation of HgS by the reaction between Hg and S (i.e., Hg+S→HgS) was thermodynamically favored with ΔG values of −50.66 kJ mol$^{-1}$ for cinnabar and −47.73 kJ mol$^{-1}$ for metacinnabar, the relatively stable organosulfur structure of PPS led to insufficient accessibility for binding mercury[36]. Mercury decomposition by the spent PPS was investigated with a temperature-programmed decomposition (TPD) experiment. Before the TPD experiments, possible interference from PPS decomposition on the measurements of the mercury analyser were first excluded (shown in Supplementary Fig. 13). Figure 6b shows that the mercury on the spent PPS decomposed at approximately 160 °C, which was much lower than the decomposition temperature for HgS crystals (α-HgS, 380 °C, and β-HgS, 295 °C)[37]. This suggested that $Hg^0$ was provisionally retained by the sulfur of PPS rather than permanently immobilized as stable HgS crystals[37]. Thus, mercury might be redischarged into the gas phase, resulting in an inferior $Hg^0$ adsorption capacity. Coincidentally, this weak adsorption of mercury by the sulfur in PPS provided continuous channels for mercury transport to the selenium ligands. The buffering role of sulfur in transporting mercury accelerated the utilization of selenium in Se/PPS. When selenium was added to PPS via the postsynthetic method, the $Hg^0$ adsorption efficiency was increased by 32.2% compared to that of pristine PPS (shown in Fig. 6a). The inadequate $Hg^0$ adsorption was probably attributable to the weakly adsorbed mercury that migrated from sulfur sites and could not be captured by the sparse and patchy selenium sites for permanent immobilization. Thus, in addition to a high binding affinity for mercury, accessibility also

played a fundamental role in aiding $Hg^0$ adsorption with enhanced transport and diffusion.

The $Hg^0$ adsorption performance of PPS was promoted by the in situ synthetic method and the construction of abundant, highly available, and active selenium ligands (shown in Fig. 6a). The solution alkalinity (i.e., NaOH concentration) during the in situ synthesis played a critical role in the surface coverage and crystal form of selenium, thus leading to the varied $Hg^0$ adsorption performance of Se/PPS-I. As shown in Fig. 6c, the $Hg^0$ adsorption capacity of Se/PPS-I without NaOH in the synthetic procedure was very similar to that of pristine PPS. This was attributed to the fact that the selenium precursor (i.e., $SeO_3^{2-}$) was difficult to reduce to elemental selenium by glutathione in a neutral environment, while $SeO_3^{2-}$ was inert to $Hg^0$, as demonstrated in Supplementary Fig. 14. This suggested that the inadequate $Hg^0$ adsorption capacity was attributable to the sulfur in PPS rather than $SeO_3^{2-}$. The addition of NaOH caused the reduction of $SeO_3^{2-}$ to elemental selenium, thus enhancing the $Hg^0$ adsorption capacity of Se/PPS-I. As shown in Supplementary Fig. 15, the sample prepared with 1% NaOH displayed much poorer $Hg^0$ adsorption than that prepared with 0.25% NaOH at a relatively low temperature (75 °C). It should be noted that the selenium amount in the reaction system was normalized by adjusting the adsorbent dosage. This suggested that in addition to the surface coverage, the crystalline form of the selenium played a crucial role in $Hg^0$ adsorption, i.e., red selenium was more active in $Hg^0$ adsorption than black selenium. The varied $Hg^0$ adsorption capacities of powdery red selenium and black selenium confirmed this interpretation (shown in Supplementary Fig. 16). Figure 6c shows that very close $Hg^0$ adsorption capacities were obtained for different Se/PPS-I samples at a relatively high temperature of 125 °C when the NaOH concentration was above 0.1%. Although different surface coverage and crystalline forms of selenium occurred on Se/PPS-I, the selenium ligands for binding $Hg^0$ were adequate for a short-term experiment, thus leading to close $Hg^0$ adsorption capacities. This confirmed the $Hg^0$ adsorption activities of the different selenium crystalline forms, since a higher temperature provided more heat to reach the activation energy needed for the reaction between mercury and black selenium.

The effects of the selenium surface coverage and availability on $Hg^0$ adsorption by Se/PPS-I were explored by adjusting the $SeO_3^{2-}$ concentration. As shown in Fig. 6d, an increase in the $SeO_3^{2-}$ concentration from 2.75% to 5.5% led to increased $Hg^0$ adsorption by Se/PPS-I, which was attributed to the increased number of selenium sites for $Hg^0$ immobilization. The highly dispersed selenium particles formed domes and dense films when the $SeO_3^{2-}$ concentration was increased to 11% and 22%, respectively. These variations in morphology were expected to reduce the accessibility of the selenium ligands. However, it was found that the $Hg^0$ adsorption capacity was very close to that of Se/PPS-I with a $SeO_3^{2-}$ concentration of 5.5%, suggesting that mercury underwent subsurface penetration to populate internal sites for immobilization[38]. This observation also explained the similar $Hg^0$ adsorption capacities of the aforementioned Se/PPS-I samples prepared with different NaOH concentrations. However, inferior $Hg^0$ adsorption performance was exhibited by Se/PPS-I with a further increase in the $SeO_3^{2-}$ concentration to 44%, and a dense film with scarce pores was generated on the sample. The thorough coverage of selenium on PPS and the high $K_a$ for Hg and Se led to firm fixation of the mercury on the Se/PPS-I outside surface rather than preadsorption on the sulfur sites[39]. This interpretation was verified by the limited $Hg^0$ adsorption capacity of selenium powder with a dense surface (shown in Fig. 6a). Thus, the diffusion of mercury into the internal selenium layer was hindered, and $Hg^0$ adsorption was inhibited.

To interpret the transport of mercury between the sulfur and selenium sites, the $Hg^0$ adsorption capacities of Se/PPS-I were tested at different temperatures. As shown in Fig. 6e, the $Hg^0$ concentration was decreased from 1000 to approximately 762 and 310 μg m$^{-3}$ at 50 and 75 °C, respectively. The inadequate $Hg^0$ adsorption by Se/PPS-I at low

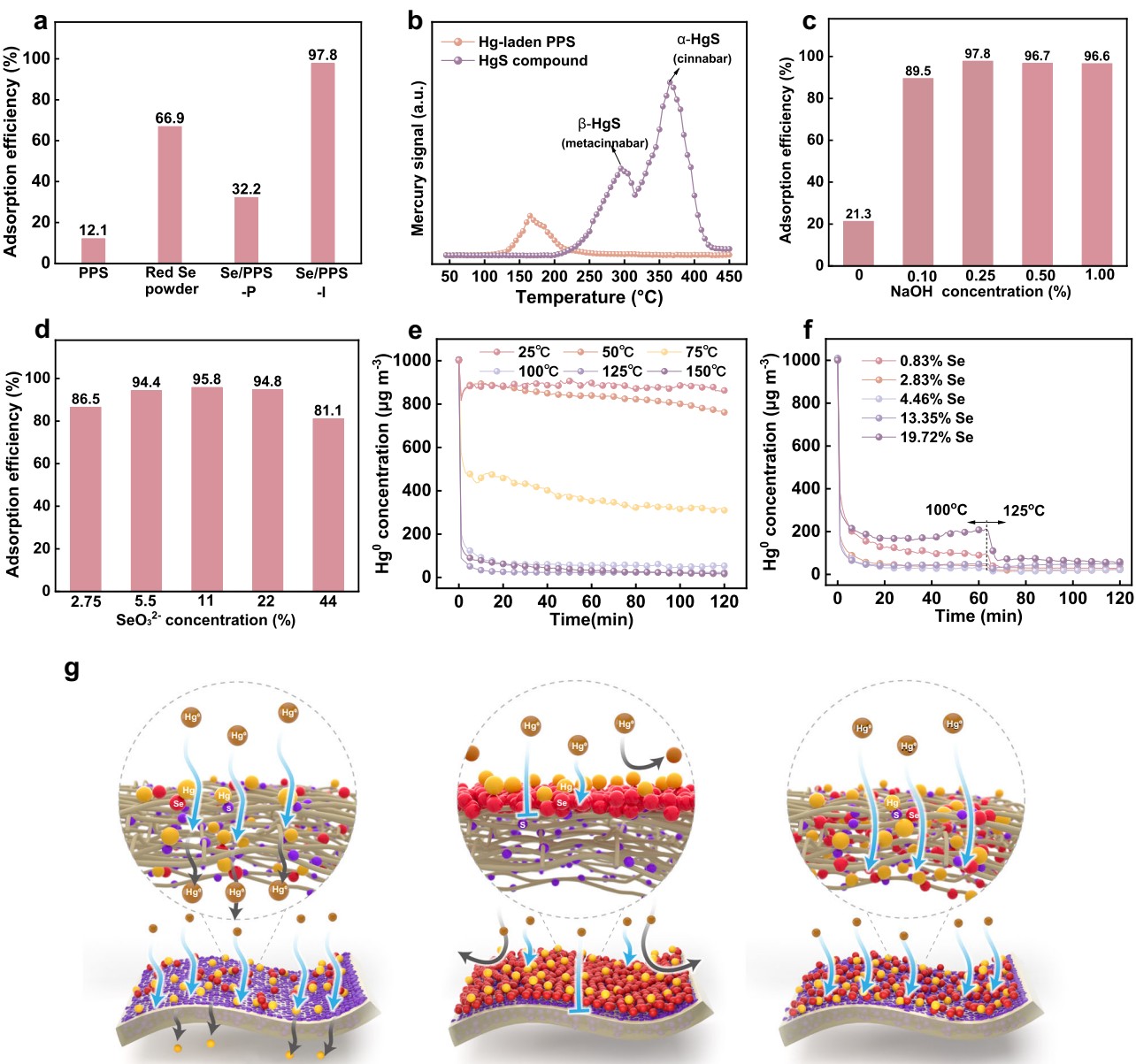

**Fig. 6 | Hg⁰ adsorption and mercury decomposition characteristics. a** Hg⁰ adsorption efficiencies of different samples, **b** mercury decomposition curves for Hg-laden PPS and HgS compound, effects of (**c**) NaOH concentration (SeO$_3^{2-}$ concentration of 5.5%, adsorption temperature of 125 °C), **d** SeO$_3^{2-}$ concentration (NaOH concentration of 0.25%, adsorption temperature of 125 °C), **e**, **f** temperature on Hg⁰ adsorption performance, and **g** Hg⁰ adsorption over different Se/PPS-I.

temperatures was ascribed to slow transport of the mercury from sulfur to selenium. As a result, the weakly adsorbed mercury might be discharged into the flue gas from Se/PPS-I and generate a low apparent adsorption efficiency. The increase in temperature to 100–150 °C increased Hg⁰ adsorption by the Se/PPS-I. Figure 6f shows that when the reaction temperature was raised from 100 to 125 °C, the amounts of Hg⁰ adsorbed by Se/PPS-I samples with different selenium surface coverages and availabilities were very close. Supplementary Fig. 17 shows that limited Hg⁰ adsorption on PPS occurred at temperatures of 50–150 °C, and the increased amounts of Hg⁰ adsorbed by the sulfur in PPS at high temperatures were insignificant. This suggested that a relatively high temperature provided more energy to accelerate mercury diffusion from sulfur to selenium for permanent immobilization.

The Hg⁰ adsorption on Se/PPS-I were fitted with typical kinetic models, the results of which are shown in Supplementary Fig. 18. The pseudofirst-order and intra-particle diffusion models fitted well with the experimental data, with correlation coefficient ($R^2$) of 0.97 and

0.99, respectively. This suggests that Hg⁰ adsorption on Se/PPS-I was controlled by external and internal mass transfer rather than the common controlling step (i.e., chemisorption) for many other materials reported in previous studies[40,41]. This unusual observation was attributed to the strong Hg-Se binding affinity and abundant chelating sites on Se/PPS-I, which increased the chemisorption of Hg⁰ on Se/PPS-I. The sulfur sites of PPS acted as buffers to provide unimpeded channels for transfer of Hg⁰ to selenium. As a result, the mass transfer resistance during the Hg⁰ adsorption process was overcome to a large extent. Figure 6g shows a schematic illustration of Hg⁰ adsorption on Se/PPS-I samples with different selenium surface coverages and availabilities. Hg⁰ was transported easily through the tangled-structured PPS with low selenium loading amounts and was weakly adsorbed at the sulfur sites. However, there was inadequate selenium to capture the mercury that migrated from sulfur, which was thus discharged into the flue gas and resulted in a low adsorption efficiency. In contrast, the added selenium thoroughly covered the PPS skeleton, which resulted

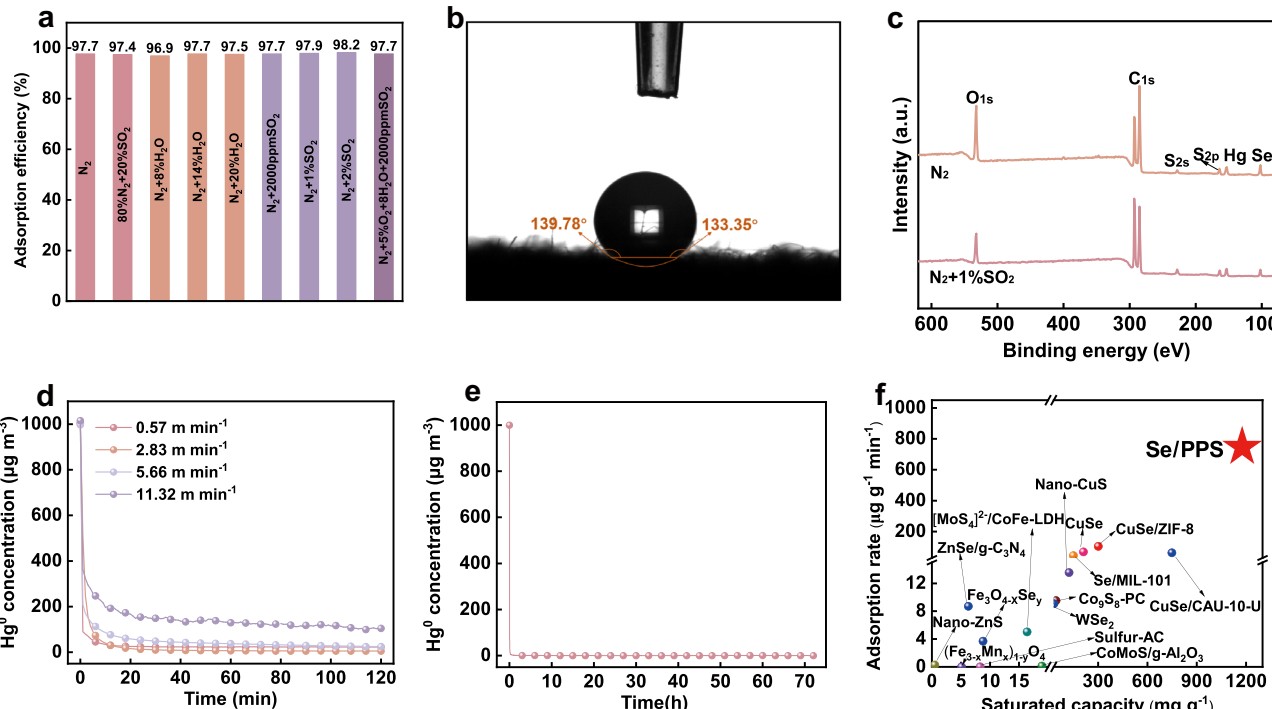

**Fig. 7 | Effect of the operating conditions on Hg⁰ adsorption by Se/PPS-I and the implications for ultralong-term industrial application. a** Hg⁰ adsorption at different atmospheres, **b** water contact angle of Se/PPS-I, **c** XPS spectra of spent Se/PPS-I at different atmospheres, **d** Hg⁰ adsorption at different GHSVs, **e** long-term stability of Se/PPS-I for Hg⁰ adsorption, **f** comparison of Hg⁰ adsorption capacities and rates for Se/PPS-I with those of other adsorbents (the corresponding data can be found in Supplementary Table 2).

in direct immobilization of the Hg⁰ on selenium sites. In this scenario, only the external selenium layer was employed for binding Hg⁰, while the diffusion of mercury to the internal selenium ligands was impeded. Thus, a moderate selenium surface coverage was required to achieve satisfactory Hg⁰ adsorption, in which adequate sulfur in the PPS was exposed to preadsorb the Hg⁰, and the weakly adsorbed mercury then migrated to the adjacent selenium ligands for permanent immobilization.

**Implications for ultralong-term use and mercury recovery**
Since Se/PPS-I was designed for use with different industrial flue gases, the Se/PPS-I should withstand harsh operating conditions. As shown in Fig. 7a, Se/PPS-I did not exhibit interference from typical flue gas components, including $H_2O$, $SO_2$ and particulate matter. The water contact angle above 130° indicated the hydrophobicity of Se/PPS-I (shown in Fig. 7b). As a result, the likelihood of $H_2O$ covering the selenium surface was diminished, thus eliminating the adverse effect of $H_2O$ on Hg⁰ adsorption. Additionally, there were no obvious changes in the oxidation states of Se and S on the fresh and pretreated Se/PPS (shown in Fig. 7c). Thus, unlike other adsorbents reported previously, such as activated carbons, metal oxides, and noble metals, which were partially deactivated by the presence of $H_2O$ and $SO_2$,[31] Se/PPS-I maintained stable Hg⁰ adsorption in flue gases containing 20% $H_2O$ and 2% $SO_2$ (shown in Supplementary Fig. 19). Additionally, as a monolithic adsorbent, Se/PPS-I was designed for use in a fixed-bed process. The excellent performance of Se/PPS-I at 100–150 °C (Fig. 6e) implied that Se/PPS-I could serve as a dust filter bag material to achieve simultaneous removal of particulate matter and Hg⁰ from flue gases, as illustrated in Supplementary Fig. 20. Supplementary Fig. 21 shows that the Hg⁰ removal capacities of Se/PPS with and without coverage by particulate matter were similar, implying feasibility for long-term use under high-concentration dust conditions. Figure 7d shows the impact of the gas flow rate on Hg⁰ adsorption by Se/PPS-I. The outlet Hg⁰ concentration was nearly 0 μg m⁻³ when the specific gas flow rate was

below 5.66 m min⁻¹, which was much higher than that in real-world applications (0.8–2.4 m min⁻¹).

A large Hg⁰ adsorption capacity was crucial for practical application of the monolithic Se/PPS-I in a fixed-bed since ultralong-term stability would prolong the service life and avoid frequent adsorbent replacement. As shown in Fig. 7e, almost no Hg⁰ was detected at the adsorbent bed outlet over 70 h with a specific gas flow rate of 1.13 m min⁻¹, even though the inlet Hg⁰ concentration was as high as 1000 μg m⁻³. The Hg⁰ adsorption capacity of Se/PPS-I was 1203.4 mg g⁻¹ when reaching 80% breakthrough (shown in Supplementary Fig. 22). Additionally, the adsorption rate was calculated as 1005.6 μg g⁻¹ min⁻¹. This Hg⁰ adsorption capacity and rate are the highest recorded values among various materials (shown in Fig. 7f and Supplementary Table 2)[14,19,42–51]. Based on the saturated Hg⁰ adsorption capacity and the adsorption breakthrough curve, the Hg⁰ concentration in coal-fired flue gas is expected to be reduced from approximately 20 μg m⁻³ to below 1.7 μg m⁻³ (i.e., the current most rigorous emission limit in the world) for 9940 h if Se/PPS-I was used with a real specific gas flow rate of 0.8 m min⁻¹. The enriched mercury on Se/PPS-I can be recovered as liquid mercury metal with a device capable of rapidly decomposing mercury compounds and efficiently separating liquid mercury (shown in Supplementary Fig. 23). Specifically, the decomposition temperature was set as 300 °C based on the TPD results, i.e., the mercury on the spent Se/PPS-I was decomposed to gaseous Hg⁰ in the temperature range 150–300 °C. The gaseous Hg⁰ was condensed to liquid mercury in an ice bath for collection (shown in Supplementary Fig. 24a). Approximately 87% mercury on the spent Se/PPS-I was recovered with this strategy. The selenium released during the decomposition of mercury was also condensed for recovery based on the variations in saturated vapor pressures (shown in Supplementary Fig. 24b). These results demonstrated that Se/PPS-I, with a large Hg⁰ adsorption capacity, a high Hg⁰ uptake rate, excellent resistance to flue gas interference, and mercury and selenium recovery capabilities, is suitable for real-world use in cleaning industrial flue gases.

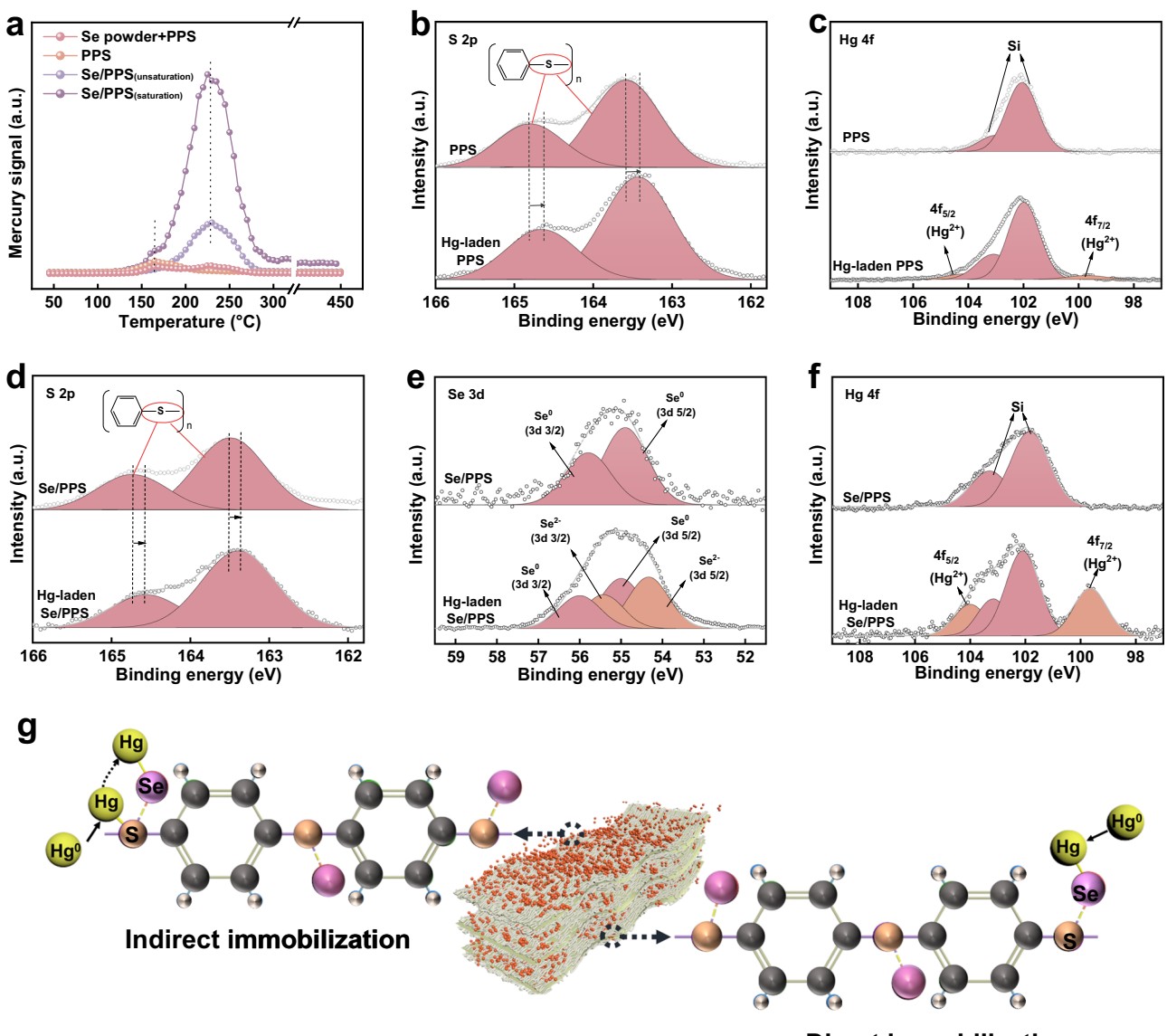

**Fig. 8 | Reaction mechanism for Hg⁰ adsorption over Se/PPS-I. a** TPD patterns for the spent pristine PPS, Se powder plus PPS, Se/PPS (unsaturation), and Se/PPS (saturation), **b** S 2p and **c** Hg 4f XPS spectra of spent PPS, **d** S 2p, **e** Se 3d and **f** Hg 4f XPS spectra for spent Se/PPS, **g** Hg⁰ adsorption pathway on Se/PPS.

## Reaction mechanism for Hg⁰ adsorption on Se/PPS-I

To clarify the role of sulfur in facilitating the immobilization of mercury, the Hg⁰ removal performances of polyester, PPS, Se/polyester, and Se/PPS were compared. The two supporters (polyester and PPS) as adopted shared similar tangled structures, while polyester contains no sulfur ligand. As shown in Supplementary Fig. 17, when raising the reaction temperature, the Hg⁰ removal performances of PPS improved, but the Hg⁰ removal performance of polyester was slightly compromised. This phenomenon indicates that Hg⁰ adsorption in PPS was primarily attributed to chemical interaction in the presence of sulfur ligands, because chemical interaction can be improved by higher reaction temperature with more energy input. However, for polyester, the physisorption effects might dominate, and higher temperature caused the desorption of physiosorbed products. The observations indicate that, in the absence of Se, Hg⁰ might chemically interact with sulfur ligands in PPS at relatively high temperature. Such chemical interaction was further supported by another controlled set of experiment between Se/polyester and Se/PPS. Se/polyester and Se/PPS exhibited comparable Hg⁰ removal performances at temperatures lower than 75 °C, which suggests that, before chemical interaction between mercury and sulfur

was adequately activated, Se/polyester and Se/PPS did not vary in Hg⁰ removal capacity. Contrarily, when the reaction temperature raised to higher than 75 °C, a temperature coincided with the activation temperature of chemical interaction, Se/PPS outperformed Se/polyester for Hg⁰ capture. This suggests the faciliting role of sulfur ligands in promoting the Hg⁰ removal on Se/PPS.

TPD experiments were conducted to confirm the migration of transient mercury-sulfur species to selenium sites for permanent immobilization. As shown in Fig. 8a, the Hg-laden selenium powder +PPS (i.e., selenium powder mechanically mixed with PPS) exhibited two characteristic decomposition peaks located at -160 °C and 230 °C. By comparing the Hg-TPD patterns of pristine PPS, unsaturated Se/PPS-I (adsorption of Hg⁰ for 2 h), and selenium powder +PPS, it was found that the characteristic peak at ∼ 160 °C was attributed to the chemical interaction between PPS-sulfur and mercury (no physisorption was accounted considering the high pretreatment and decomposition temperature), and the one centered at ∼ 230 °C was ascribed to HgSe[12]. It was found that no characteristic peak accounting for mercury-PPS interaction was observed in unsaturated Se/PPS-I because, when the sample was far from being saturated, and the

amount of active selenium was abundant in the sample, mercury interacted with PPS supporter, if any, would be immediately transferred to mercury selenide on the surface of Se/PPS-I. However, if increased the pretreatment time and consumed most of the active selenium sites in Se/PPS-I, the interaction between PPS and mercury could be observed, which was manifested by the occurrence of 160 °C peak in the saturated Se/PPS-I (adsorption of Hg$^0$ for more than 50 h). This further supports that the PPS-sulfur did interact with mercury, and the absence of mercury-sulfur intermediates in unsaturated Se/PPS is mainly ascribed to the spontaneous and rapid transformation of the intermediates into the mercury selenide final product when selenium sites were abundant. Thus, it was speculated that the plentiful sulfur in Se/PPS-I provided bridges to intercept Hg$^0$, and the weakly adsorbed mercury subsequently migrated to the selenium interfaces for permanent immobilization.

The surface chemistries of fresh and Hg-laden PPS and Se/PPS-I were determined with XPS to investigate the roles of sulfur and selenium in Hg$^0$ adsorption. The C 1s and O 1s spectra of PPS and Se/PPS-I changed insignificantly after adsorbing Hg$^0$, suggesting that C and O in PPS did not participate in Hg$^0$ adsorption. As shown in Fig. 8b, the S 2p binding energies of fresh PPS were 163.6 and 164.8 eV, which were consistent with those generally found for organosulfur compounds. In contrast, after adsorbing Hg$^0$, the S 2p binding energies of the Hg-laden PPS were shifted to 163.4 and 164.6 eV. The negative shifts implied interactions between the sulfur and Hg$^0$ in PPS, i.e., Hg$^0$ donated electrons to sulfur, leading to the oxidation of Hg$^0$ and electron-rich sulfur atoms. The Hg 4f spectrum showed a doublet resulting from the 4f$_{7/2}$ and 4f$_{5/2}$ peaks at 99.7 and 104.1 eV, respectively (shown in Fig. 8c), again demonstrating the formation of Hg$^{2+}$. The dominant peak at 102.0 eV was attributed to silicon introduced from the quartz wool during Hg$^0$ adsorption experiments rather than mercury. Negative shifts of the S 2p binding energies for Hg-laden Se/PPS-I are also shown in Fig. 8d, suggesting that the sulfur in Se/PPS-I still participated in Hg$^0$ adsorption. As shown in Fig. 8e, compared to fresh Se/PPS-I, two peaks located at 54.4 and 55.5 eV were additionally detected in the Se 3d spectrum for Hg-laden Se/PPS-I, which were consistent with the Se 3d binding energies of HgSe[52]. Additionally, two well-resolved peaks corresponding to HgSe were observed at 99.6 and 104.0 eV in the Hg 4f spectrum of Hg-laden Se/PPS-I (shown in Fig. 8f)[52]. However, no additional peaks corresponding to HgS or other mercury species (e.g., Hg$^0$, HgO) were observed in the Hg 4f spectrum. This demonstrated the migration of mercury from sulfur to selenium and the formation of stable HgSe (indirect immobilization). As shown in Supplementary Fig. 25, the peaks corresponding to the Se$^{2-}$ ions in HgSe were detected in the Se 3d spectrum of Hg-laden Se/polyester. Additionally, HgSe was also detected in the Hg 4f spectrum. This suggested that Hg$^0$ was directly immobilized on the selenium sites without the participation of sulfur, which was ascribed to the higher binding affinity of mercury (direct immobilization)[53]. In summary, the accessibility of selenium was enhanced via two pathways, as illustrated in Fig. 8g, thus overcoming the transfer limitations caused by the unfavorable pore structure and the short resistance time.

In summary, this work demonstrated for the feasibility of enhancing Hg$^0$ adsorption on functionalized substrates via a biomimetic pathway. The population density, distribution, and crystalline form of selenium were rationally regulated by adjusting the concentrations of NaOH and SeO$_3^{2-}$ from 0% to 1% and 2.75% to 44%, respectively. At a reaction temperature higher than 100 °C, which was within the operation temperature range for fabric fibers under practical flue gas cleaning scenarios, the resultant Se/PPS-I displayed a Hg$^0$ adsorption capacity and uptake rate of 1621.9 mg g$^{-1}$ and 1005.6 μg g$^{-1}$ min$^{-1}$, respectively. The excellent Hg$^0$ adsorption performance of Se/PPS-I was attributed to the plentiful sulfur sites in PPS that served as buffers for Hg$^0$ transport to the adjacent selenium. The resistance of Se/PPS-I to flue gas interference enables ultralong-term use under harsh flue gas conditions, and it is expected to serve for approximately 10,000 h

without changing the adsorbent. This work developed an effective Hg$^0$ adsorbent and provided guidance for biomimetic design of advanced functional filters for pollutant abatement.

## Methods

### Materials
The raw materials used in this work including glutathione (GSH, reduced form, C$_{10}$H$_{17}$N$_3$O$_6$S, 98%), sodium hydroxide (NaOH, 98%), nitric acid (HNO$_3$, 70%), hydrochloric acid (HCl, 36.5%) were purchased from Aladdin. Commercial polyphenylene sulfide (PPS, (C$_6$H$_4$S)$_n$) and polyester ((C$_{10}$H$_8$O$_4$)$_n$) were purchased from Alibaba.Sodium selenite (Na$_2$SeO$_3$, 97%) was purchased from Sinopharm.

### Sample preparation
The in-situ synthesis procedure for Se/PPS was as follows. First, 2.6 mmol sodium selenite (Na$_2$SeO$_3$) was fully dissolved in deionized water and stirred for 0.5 h. Then, a piece of commercially purchased PPS fabrics (6 g) was immersed in the solution and oscillated for 6 h continuously. After that, 5.85 mmol glutathione (GSH, reduced form) was added into the solution and oscillated for another 6 h. Then, 12.5 mmol sodium hydroxide (NaOH) solution was dropwise added into the suspension to bring the pH above 12 and oscillated for 5 h at room temperature. Finally, the product was rinsed with deionized water before drying in vacuum at 110 °C for 12 h. The as-prepared sample is denoted as Se/PPS-I. The dosage of selenium precursor and NaOH was also adjusted to synthesize Se/PPS-I samples with various Se coating mounts and morphologies. Polyester fabric without containing sulfur was also adopted to be functionalized by the in-situ synthesis method (denoted as Se/Polyester). In addition, the SeO$_3^{2-}$ loaded PPS (denoted as SeO$_3^{2-}$/PPS) was prepared by the same procedure for obtaining Se/PPS-I but without adding GSH and NaOH.

As a reference, the post synthesis method was adopted to prepare Se/PPS-P. First, a suspension containing selenium particles was prepared by dissolving 2.6 mmol of Na$_2$SeO$_3$, 5.85 mmol of GSH, and 12.5 mmol of NaOH in deionized water and stirring for 5 h. Then, a piece of PPS fabric (6 g) was immersed in the solution and oscillated for 5 h continuously. Finally, the product was rinsed with deionized water several times and dried in vacuum at 110 °C for 12 h. The dosages of selenium precursor and NaOH were the same as those for preparing Se/PPS-I samples.

The powdery selenium preparation procedure was same as that for Se/PPS but only without containing PPS. Specifically, 2.6 mmol Na$_2$SeO$_3$ was fully dissolved in deionized water and stirred for 0.5 h. After that, 5.85 mmol GSH (reduced form) was added into the solution and oscillated for another 6 h. Then, 12.5 mmol NaOH solution was dropwise added into the suspension to bring the pH above 12 and oscillated for 5 h at room temperature. Finally, the powdery selenium sample was obtained after centrifuging, rinsing with deionized water, and drying in vacuum at 110 °C for 12 h.

### Sample characterization
The thermal stability was determined by thermogravimetric analysis (TG-DTA, NETZSCH STA 2500), which was performed from 50 to 800 °C at 10 °C min$^{-1}$. Pure nitrogen (N$_2$) was used as the carrier gas for TG analysis, the flow rate of which was 50 mL min$^{-1}$. The crystallinity of the sample was determined by X-ray Diffraction (XRD, D8 Bruker AXS) using a Cu-Kα radiation source (40 kV, 40 m A$^{-1}$). The morphology and elemental mapping of the sample were studied by a scanning electronic microscope (SEM, FEI F50) equipped with an energy-dispersive X-ray (EDX). The valance states of each element on the sample were investigated by X-ray photoelectron spectroscopy (XPS). The binding energy value of C 1s (284.8 eV) was used as a reference for spectral correction. Transmission electron microscope (TEM, EOL JEM 2100 F microscope) and high-resolution TEM (HRTEM) were used to study the morphology and structure of the sample. The water contact angle was tested by a water contact angle detector (Biolin Theta Flex). The Brunauer–Emmett–Teller

(BET) surface area of the sorbents was determined by the $N_2$ adsorption and desorption method with a BET analyzer (ASAP 2460, Micromeritics, USA). The selenium amount on the sample was tested by inductively coupled plasma mass spectrometry (ICP-MS, Agilent 7800) after digested. Firstly, 0.2 g sample, 3 mL de-ionized water and 3 mL aqua regia were put in an extraction bottle, which was kept at 75 °C until the sample was dissolved completely. Then, the solution was acid-driving treated, diluted by de-ionized water and added $HNO_3$ to maintain the pH value below 2. After that, the selenium content in the solution was measured by an ICP-MS. The stability of selenium on the sample during long-term application was also tested 1 L min$^{-1}$ $N_2$ for 14 d, and then the selenium of the sample was tested by ICP-MS. The spent sample was characterized by XPS to investigate the surface chemistry variation. The fresh sample was pretreated by 1 L min$^{-1}$ $N_2$ containing 1000 μg m$^{-3}$ Hg$^0$ at 125 °C for 2 or 50 h and purging by 1 L min$^{-1}$ $N_2$ to remove the unstable mercury to obtain a mercury-laden sample. The mercury adsorption product on the spent sorbent was studied by temperature programmed desorption (TPD) experiments, where the mercury species could be identified by comparing the desorption characteristics with that of the reference pure mercury compounds. The sample adopting for the TPD experiment was same with that for XPS analysis. The TPD experiment was conducted from 40 to 450 °C with a heating rate of 5 °C min$^{-1}$ to decompose the mercury adsorbed on the spent sorbent into Hg$^0$, and the Hg$^0$ was carried by $N_2$ (500 mL min$^{-1}$) into the mercury analyzer for detection.

## Gas phase Hg$^0$ adsorption tests

The Hg$^0$ adsorption performance was studied in a fixed-bed adsorption system. The flue gases were fed by cylinders, with a total flow rate of 1 L min$^{-1}$. Hg$^0$ vapor was supplied by a Hg$^0$ permeation tube, which was maintained at 90 °C to obtain a stable Hg$^0$ concentration of 1000 μg m$^{-3}$. The Hg$^0$ vapor was introduced into the reactor by $N_2$ (0.3 L min$^{-1}$). The Hg$^0$ concentration was measured by an online mercury analyzer (Lumex, RA-915M). Moisture was provided by an impinger containing de-ionized water which was heated at 80 °C and delivered by 0.08 L min$^{-1}$ $N_2$ into the reactor. The temperature of the reactor was controlled by a tubular furnace. Eight sets of experiments were carried out. The experimental conditions are summarized in Supplementary Table 1. Set I and II experiments were conducted to investigate the effects of synthesis conditions (i.e., selenium coating amount and NaOH adding amount) on the Hg$^0$ removal performances of Se/PPS. In Set III and Set VII experiments, the Hg$^0$ removal performances of Se/PPS under different operation conditions were studied. First, the Hg$^0$ removal performances of Se/PPS at different reaction temperatures (25, 50, 75, 100, 125, and 150 °C) were tested in Set III experiment. Then, the applicability of the Se/PPS for Hg$^0$ removal under different atmospheres was investigated in Set IV experiment. In this set of experiments, the effects of $SO_2$ and $H_2O$ on the Hg$^0$ adsorption performance of Se/PPS were confirmed by conducting the Hg$^0$ adsorption test under high $SO_2$ and $H_2O$ concentrations of 2% and 20%, respectively. As a comparison, powdery selenium was adopted for Hg$^0$ adsorption test to justify the superior performance of Se/PPS in removing Hg$^0$. The dosage of powdery selenium adopted for Hg$^0$ adsorption test was 3.1 mg, which was same as the amount of selenium on Se/PPS-I. In Set V experiments, the effects of particulate matter on the Hg$^0$ adsorption performances of Se/PPS were investigated. The Se/PPS was firstly covered by fly ash collected from a typical coal-fired power plant, and the Hg$^0$ adsorption performance was then tested and compared with that of pristine Se/PPS. In Set VI experiments, the effect of specific gas flow rate on the Hg$^0$ adsorption performance of Se/PPS was studied. In Set VII experiments, the stability of Hg$^0$ adsorption as well as the saturation capacity was studied. In Set VIII experiments, the Hg$^0$ adsorption performances of polyester and Se/Polyester with a similar tangled structure but without sulfur were studied as references. In each set of experiments, the dosage of the sample was 1.8 ×15 mm (H×D) except specifically noted. The Hg$^0$ adsorption capacity and Hg$^0$ removal efficiency were calculated by the following equations.

$$Q_t = \frac{1}{m} \int_{t_1}^{t_2} (C_{in} - C_{out}) \times f \times dt \quad (1)$$

$$\eta = \frac{\int_{t_1}^{t_2} (C_{in} - C_{out}) \times f \times dt}{\int_{t_1}^{t_2} C_{in} \times f \times dt} \quad (2)$$

where $Q_t$ is the Hg$^0$ adsorption capacity (g Hg g$^{-1}$ adsorbent), $\eta$ is the Hg$^0$ removal efficiency (%), $m$ is the Se mass of sorbent (g), $f$ is the gas flow rate (m$^3$ h$^{-1}$), $t$ is the accumulated time of every set experiment (h) and $t = 120$ min when calculating the adsorption efficiency, $C_{in}$ and $C_{out}$ is the inlet and outlet Hg$^0$ concentration (μg m$^{-3}$).

## Mercury and selenium recovery

The recovery of mercury from spent Se/PPS was conducted using a device capable of rapidly decomposing mercury adsorption products to gaseous Hg$^0$, which was then condensed to liquid mercury for collection (as illustrated by Supplementary Fig. 23). The decomposition temperature was determined by a TPD experiment, i.e., the temperature which could guarantee the full decomposition of mercury from spent Se/PPS. The spent Se/PPS was placed at a furnace to decompose the mercury adsorption products (i.e., HgSe), and the gaseous Hg$^0$ was condensed to liquid mercury in an ice-bath. Meanwhile, the selenium on Se/PPS would be released accompanying with the decomposition of HgSe. Thus, both mercury and selenium were condensed for recovery based on their different saturated vapor pressure.

## Description of adsorption kinetic models

The Pseudo-first-order model, Pseudo-second-order model, Intra-particle diffusion model, and Elovich model were adopted for simulating the Hg$^0$ adsorption process. The Pseudo-first-order model is based on the mass balance. The Hg adsorption rate was proportional to the difference between the equilibrium capacity and the adsorbed amount at any time, as described following:

$$\frac{dq_t}{dt} = k_1 (q_e - q_t) \quad (3)$$

Equation (3) could be modified to the following equation based on the initial conditions of $t = 0$ $q_t = 0$ and $t = t$ $q_t = q_t$:

$$q_t = q_e \left(1 - e^{-k_1 t}\right) \quad (4)$$

where $q_t$ and $q_e$ represent the adsorbed mercury amount at any time $t$ and equilibrium time (μg·g$^{-1}$). $k_1$ represents the rate constant (min$^{-1}$). $q_e$ and $k_1$ can be attained by fitting the adsorption curve.

The pseudo-second-order model represents that the surface diffusivity is inversely proportional to the square of concentration variation on sorbent surface, which could be described as following:

$$\frac{dq_t}{dt} = k_2 (q_e - q_t)^2 \quad (5)$$

Equation (5) can be modified to the following equation based on the initial conditions of $t = 0$ $q_t = 0$ and $t = t$ $q_t = q_t$:

$$\frac{t}{q_t} = \frac{1}{k_2 q_e^2} + \frac{1}{q_e} t \quad (6)$$

where $k_2$ represents the rate constant (min$^{-1}$). The term $k_2 q_e^2$ was the initial adsorption rate.

The Intra-particle diffusion model assumes that the intra-particle diffusivity is constant and the diffusion direction is radial. The model

can be interpreted by the following equation:

$$q_t = k_{id}t^{0.5} + C \qquad (7)$$

where $k_{id}$ represents the intraparticle diffusion rate constant, C is proportional to the boundary layer.

The Elovich model assumes that sorption takes place in two phases, i.e., a fast initial reaction associated with the movement of the sorbate to external sites, and a slower diffusion in and out of the microspores over sorbent. This model can be described by the following equation:

$$\frac{dq_t}{dt} = \alpha \exp(-\beta q_t) \qquad (8)$$

where $\alpha$ represents the initial rate, $\beta$ is related to the extent of surface coverage and activation energy for chemisorption. If $t$ is much larger than $t_0$ this equation is modified as follows:

$$q_t = \frac{1}{\beta}\ln(\alpha\beta) + \left(\frac{1}{\beta}\right)\ln t \qquad (9)$$

### Quantum chemical simulations

All the quantum chemical simulations were performed by using Gaussian 16 package. The exchange-correlation energy was calculated by M06-2X hybrid functional. Def2−DZVP and Def2−TZVP basis sets were applied to geometry optimization and single-point energy calculations, respectively. In addition, the DFT-D3 method was used to describe dispersion force. The electrostatic interaction analyses were conducted by using the Multiwfn package, and the related Fig. 2g was drawn by using the VMD package. The atomic coordinates of the optimized computational models refer to Supplementary Data.

### Statistics and reproducibility

For all experiments, the data were sampled according to the minimal number of independent replicates that significantly identified an effect (repeating at least three times). Repeated measurements of the evolving quantities showed deviations of less than 10% confirming reproducibility of the reported experiments.The error bar(s) are defined as the difference between the maximum(max(s)) and minimum (min(s)) values for different test rounds.

## Data availability

The data that supports the findings of the study are included in the main text and supplementary information files. Source data are provided with this manuscript. Source data are provided with this paper.

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

## Acknowledgements

This project was supported by National Key Research and Development Program of China (2022YFC3701505-04-H.L.), National Natural Science Foundation of China (52276145- J.Y., 52276144-H.L.), Natural Science Foundation of Hunan Province, China (2021JJ30851-J.Y., 2022JJ20072–J.Y.), Science and Technology Innovation Program of Hunan Province (2021RC4005-H.L., 2021GK1210-H.L.).

## Author contributions

J.Y., H.L., F.M. conceived and designed the research. H.L. supervised the work and reviewed the manuscript. J.Y., F.M. wrote and revised the manuscript. F.M., P.Z. conducted the experiments. H.Z., W.Q. conducted the quantum chemical simulations. Z.Y.participated in characterizations and data analyses. All authors discussed the results and commented on the manuscript.

## Competing interests

The authors declare no competing interests.
