## [Peer Review File · Nature Communications]

Biomimetic Mercury Immobilization by Selenium Functionalized Polyphenylene Sulfide FabricREVIEWER COMMENTS

Reviewer #1 (Remarks to the Author):

Decontamination of mercury pollution from industrial flue gases is a serious environmental, which is an urgent need to solve. This work presented biomimetic mercury immobilization by selenium functionalized polyphenylene sulfide fabric. Based on the novelty and applicability of bionic mindset, the publication of it might expand a new direction to develop efficient adsorbents in diverse environmental remediation including mercury pollution. Thus, I strongly recommend to publish this manuscript on Nature Communications after minor revisions.

1. A general introduction to the field and more detailed background specific to the research should be provided in the Abstract.
2. It is suggested to first discuss the characterizations of sorbents as prepared through in-situ and post synthesis methods before providing a diagrammatic illustration on this process. Thus, Figure 2 may be better placed after Figure 5 to serve this purpose.
3. TPD experiments were conducted to investigate the mercury species on the spent sample based on the decomposition characteristics of mercury compounds. Besides the decomposition of mercury compounds, the organic PPS fabric filter will be decomposed as well. Whether the decomposition of PPS would affect the determination of mercury by the mercury analyzer or cause confusion for investigating mercury species?
4. The authors stated that, besides the surface coverage of selenium, the crystal form of selenium played a crucial role in mercury adsorption, i.e., red selenium was more efficient than black selenium. Although the selenium amount was normalized by adjusting the adsorbent dosage to compare the mercury adsorption performances of different samples, the effect of surface distribution cannot be excluded. I suggest the authors could add experiments to compare the mercury adsorption performance of powdery red selenium and black selenium to further support the above interpretation.
5. The Se/PPS displayed a mercury recovery feature owing to its ultra-high mercury adsorption capacity. However, the authors did not explain how mercury was recovered from the Se/PPS, and what is the mercury recovery ratio?
6. The authors stated that the Se/PPS displayed ultralong-term real application potential. So, the mercury adsorption curve of Se/PPS should be provided, and the service lifetime when adopting in a fixed bed should be evaluated.
7. Whether the selenium on Se/PPS would be easily to fall off by flue gas during long-term application, since the fall off of selenium not only affecting the mercury removal performance but also cause selenium pollution?
8. Whether the particulate matter in flue gas would cover the active sites (i.e., selenium), thus hindering the mercury removal performance of Se/PPS?
9. The authors adopted the pseudo-first-order and pseudo-second-order kinetic models to simulate the mercury adsorption behavior, but the kinetic models were not present in the manuscript.

Reviewer #2 (Remarks to the Author):

A brand-new method was developed in this work to achieve the efficient capture of HgO from industrial flue gas. The most impressive novelty of this work is that it mimics the HgO detoxification mechanism in human body, which has hardly been proposed in previous studies. The novelty of this work thus makes it suitable to be published on Nature Communications. I believe it can significantly extend the research horizons and raise wide research interests in related fields. The results are somehow interesting, but there are some notes that the authors have to pay attention. The work is recommended to publish after a minor revision. Specific suggestions are as follows.

- (1) Please provide experimental details. For instance, how to obtain the spent sample for XPS and TPD tests? The sorbent dosage, flue gas atmosphere and flue gas temperature should be also described clearly for each set experiment, since the HgO adsorption efficiency was dependent on these experimental parameters.
- (2) DTA curves should be added accompany with TG curve to better illustrate the thermal stability of the Se/PPS.
- (3) The Se/PPS was designed to be adopted in a fixed-bed reactor like filter bag type dust collector to achieve simultaneous removal of particulate matters and mercury from flue gas. A scheme for HgO removal by Se/PPS was suggested to be provided.
- (4) A comparative table should be provided to demonstrate the superior performance of Se/PPS compared to other sulfur/selenium functionalized substrates in removing HgO, further proving the advanced nature of the biomimetic approach.
- (5) At the current stage, the preparation parameters of the adsorbents are not clear. The author must specify the dosage of each precursor in the adsorbent preparation section to ensure the reproducibility of this work.
- (6) The characterization of the adsorbent can be moved to the main content of the manuscript instead of placing it in the supplementary information. Additionally, specific conditions used to obtain characterization results can be further specified to support the reproducibility of this work.
- (7) In this work, the characterization of adsorbents for HgO removal performance seems to be primarily based on the HgO removal efficiency and HgO adsorption capacity presented in this study. However, the authors only provide the calculation method for HgO adsorption capacity. Please add the calculation method for HgO removal efficiency.
- (8) Due to the ability of sulfur in PPS to adsorb HgO, it is unclear whether there is still mercury sulfide present after adsorption saturation. I suggest that the authors further investigate the mercury species on Se/PPS with depleted selenium sites through long-term mercury pretreatment to firmly establish the role of sulfur in facilitating the immobilization of mercury on selenium through transport proteins.
- (9) The conclusion section mainly provides some general comments on this work. Please provide more quantitative descriptions of the main findings of this study in the conclusion section.

Reviewer #3 (Remarks to the Author):

In this work, the authors proposed a new strategy for mercury decontamination from industrial flue gas, a serious task for public health and environmental societies. The results and the mechanism in this study are reasonable. The manuscript can be accepted after revision. Some comments are listed as follows:

1. Please avoid the use of exaggerated words like “significantly”, “excellent”, “obvious”, etc. because the Nature Communications is a journal valuing scientific expressions.
2. The Abstract mentioned that the Hg₀ adsorption capacity and adsorption rate of the Se/PPS are record-breaking high values. More data from previous literatures should be included to thoroughly demonstrate the superiority of the resultant Se/PPS. Moreover, the test conditions including sorbent dosage, Hg₀ concentration and test time should be presented when comparing the Hg₀ adsorption capacity and adsorption rate.
3. The powdery selenium was adopted as a reference to compare the Hg₀ adsorption efficiency with Se/PPS. Whether the powdery selenium was a commercial sample? If so, the powdery selenium prepared with the same method adopted in this work should be prepared and the Hg₀ adsorption efficiency of it should be tested.
4. What is the dosage of powdery selenium adopted for Hg₀ adsorption test (shown in Figure 6A), and what is the level compared with the amount of selenium on Se/PPS? It should be noteworthy that the samples were in different forms (i.e., powder and monolith). Thus, same amount of selenium should be adopted when comparing the Hg₀ adsorption performance.
5. The Se/PPS displayed a mercury recovery feature owing to its great Hg₀ enrichment capacity. Mercury was recovered by decomposing the mercury adsorption products (i.e., HgSe) at high temperature. What is the decomposition temperature for mercury recovery. The volatile selenium would be released accompanying with the decomposition of HgSe at high temperature demonstrating by the TG results. Thus, how to dispose the released selenium since it is also an atmosphere pollutant.
6. The stable Hg₀ adsorption performances under harsh conditions were crucial for real-world application. The authors stated that the Se/PPS displayed excellent anti-interference ability to general detrimental flue gas components (e.g., SO₂ and H₂O). The detrimental role of these flue gas components in Hg₀ adsorption might be indeed inapparent during a short experiment time. However, the material was designed to be adopted in a fix-bed reactor, the service time would be generally several months. Thus, I commend the authors could conduct the Hg₀ adsorption test with high SO₂ and H₂O concentration to further solid the above statement.
7. The pseudo-first and pseudo-second kinetic models were adopted to simulate the Hg₀ adsorption behaviors over Se/PPS. Other kinetic models like intra-particle diffusion model and elovich model were recommended to be adopted to further investigate the Hg₀ adsorption process.
8. The mercury decomposition characteristic of Hg-laden PPS investigated by TPD experiment was described in lines Hg-TPD product of Hg desorption on PPS has been described earlier in lines 212-217, but the results were described again in the subsequent reaction mechanism analyses. Thus, it is suggested to reorganize the relevant discussion.

Detailed Response to Reviewers

Reviewer #1

Decontamination of mercury pollution from industrial flue gases is a serious environmental, which is an urgent need to solve. This work presented biomimetic mercury immobilization by selenium functionalized polyphenylene sulfide fabric. Based on the novelty and applicability of bionic mindset, the publication of it might expand a new direction to develop efficient adsorbents in diverse environmental remediation including mercury pollution. Thus, I strongly recommend to publish this manuscript on Nature Communications after minor revisions.

Comment 1:

A general introduction to the field and more detailed background specific to the research should be provided in the Abstract.

We thank the reviewer for this insightful suggestion. A sentence offering a general introduction to the field and a sentence with more detailed background specific to the research has been added to the Abstract (please refer to the Abstract section in page 2). The same sentences are shown below for your reference.

‘Highly efficient decontamination of elemental mercury (Hg^0) remains an enormous challenge for public health and ecosystem protection. The artificial conversion of Hg^0 into mercury chalcogenides could achieve Hg^0 detoxification and close the global mercury cycle.’

Comment 2:

It is suggested to first discuss the characterizations of sorbents as prepared through in-situ and post

synthesis methods before providing a diagrammatic illustration on this process. Thus, Figure 2 may be better placed after Figure 5 to serve this purpose.

We thank the reviewer for this valuable suggestion. Fig. 2 has been placed after Fig. 5 to diagrammatically illustrate the assembly process and structural properties of Se/PPS after discussing the sample characterizations. Related discussions have been correspondingly revised (please refer to the 2nd paragraph in page 9 and 1st paragraph in page 10). The same revised discussion is shown below for your reference.

‘Based on the above characterization data, the formation of Se/PPS-P and Se/PPS-I is illustrated in Fig. 5. The post synthetic process was started by coating a suspension of selenium particles, and deposition on PPS was realized via random collisions (Fig. 5a). As a result, only sparse and patchy selenium particles appeared on the Se/PPS prepared by the post synthetic method. In this scenario, even if Hg^0 was weakly adsorbed on the PPS, there were insufficient selenium ligands to bind mercury that migrated from the sulfur sites. The weakly adsorbed mercury might be discharged from the PPS into the flue gas. The *in situ* synthetic method (Fig. 5b) overcame these drawbacks, and SeO_3^{2-} was first anchored on PPS to serve as a selenium precursor, and then was reduced to elemental selenium by a reductant (i.e., GSH) under alkaline conditions. The electrostatically directed assembly of selenium on PPS was realized due to the adverse ESP distribution derived from well-organized adsorption of SeO_3^{2-} onto the PPS. Rational adjustment of the population density and the selenium distribution was achieved by adjusting the SeO_3^{2-} and NaOH concentrations (Fig. 5c), thus providing adequate selenium for binding mercury.’

Comment 3:

TPD experiments were conducted to investigate the mercury species on the spent sample based on

the decomposition characteristics of mercury compounds. Besides the decomposition of mercury compounds, the organic PPS fabric filter will be decomposed as well. Whether the decomposition of PPS would affect the determination of mercury by the mercury analyzer or cause confusion for investigating mercury species?

We thank the reviewer for this great comment. The decomposition characteristics of pure HgSe and HgS compounds mixing with PPS (without loading mercury) have been added in the revised supplementary materials and denoted as Supplemental Fig. 13 (the same figure is shown below for your reference). As shown, the mercury decomposition characteristics were very close regardless of including PPS. This demonstrates that the decomposition of PPS did not cause confusion in investigating the mercury species on the spent Se/PPS. Additionally, the mercury analyzer ran steadily when including PPS decomposition components in the carrier gas. Thus, possible interference from PPS decomposition on the measurements of the mercury analyser have been first excluded before the TPD experiments. Related discussions have been added in the revised manuscript (please refer to the 2nd paragraph on page 10).

TPD spectrum of HgSe and HgS compounds mixing with and without PPS

Stability of mercury analyzer with PPS decomposition components in the carrier gas

Comment 4:

The authors stated that, besides the surface coverage of selenium, the crystal form of selenium played a crucial role in mercury adsorption, i.e., red selenium was more efficient than black selenium. Although the selenium amount was normalized by adjusting the adsorbent dosage to compare the mercury adsorption performances of different samples, the effect of surface distribution cannot be excluded. I suggest the authors could add experiments to compare the mercury adsorption performance of powdery red selenium and black selenium to further support the above interpretation.

We thank the reviewer for this constructive suggestion. We have added experiments to compare the mercury adsorption performance of powdery red selenium and black selenium to further demonstrate

the variation of different selenium crystal forms in Hg^0 adsorption. In this scenario, the interference of selenium surface distribution on Se/PPS was excluded. The results have been added in the revised supplementary materials and denoted as Supplemental Fig. 16 (the same figure is shown below for your reference). The experimental results confirmed the role of crystal form of selenium in Hg^0 adsorption, i.e., the red selenium was more active than black selenium in Hg^0 adsorption. Related discussions on this observation have been added in the 2nd paragraph on page 11. The same revised discussion is shown below for your reference.

‘This suggested that in addition to the surface coverage, the crystalline form of the selenium played a crucial role in Hg^0 adsorption, i.e., red selenium was more active in Hg^0 adsorption than black selenium. The varied Hg^0 adsorption capacities of powdery red selenium and black selenium confirmed this interpretation (shown in Supplemental Fig. 16).’

Hg⁰ adsorption performances of powdery red and black selenium

Comment 5:

The Se/PPS displayed a mercury recovery feature owing to its ultra-high mercury adsorption capacity. However, the authors did not explain how mercury was recovered from the Se/PPS, and what is the mercury recovery ratio?

We thank the reviewer for this valuable comment. The recovery of mercury from spent Se/PPS was conducted by thermal decomposition and condensing collection, which was realized on a device

capable of rapidly decomposing mercury adsorption products to gaseous Hg^0 and then condensed to liquid mercury for collection. To better illustrate the mercury recovery process, the schematic diagram of mercury recovery system has been added in the revised supplementary materials and denoted as Supplemental Fig. 23 (the same figure is shown below for your reference). The decomposition temperature was set as 300 °C in this work, which was determined by the TPD experimental result, i.e., the mercury on the spent Se/PPS was decomposed from 150 to 300 °C according to the TPD curve. The spent Se/PPS was placed at a furnace maintaining at 300 °C to decompose the mercury into gaseous Hg^0 . The released Hg^0 was condensed to liquid mercury in an ice-bath for collection. About 87% mercury on the spent Se/PPS can be recovered by such a strategy. Related experimental methods and discussions have been added in the revised manuscript (the 2nd paragraph on page 15 and the 1st paragraph on page 16) and the revised supplementary materials (the 1st paragraph on page S6). The same revised discussion is shown below for your reference.

‘Mercury and selenium recovery

The recovery of mercury from spent Se/PPS was conducted using a device capable of rapidly decomposing mercury adsorption products to gaseous Hg^0 , which was then condensed to liquid mercury for collection (as illustrated by Supplemental Fig. 23). The decomposition temperature was determined by a TPD experiment, i.e., the temperature which could guarantee the full decomposition of mercury from spent Se/PPS. The spent Se/PPS was placed at a furnace to decompose the mercury adsorption products (i.e., HgSe), and the gaseous Hg^0 was condensed to liquid mercury in an ice-bath. Meanwhile, the selenium on Se/PPS would be released accompanying with the decomposition of HgSe . Thus, both mercury and selenium were condensed for recovery based on their different saturated vapor pressure.’

‘The enriched mercury on Se/PPS-I can be recovered as liquid mercury metal with a device capable

of rapidly decomposing mercury compounds and efficiently separating liquid mercury (shown in Supplemental Fig. 23). Specifically, the decomposition temperature was set as 300 °C based on the TPD results, i.e., the mercury on the spent Se/PPS-I was decomposed to gaseous Hg^0 in the temperature range 150–300 °C. The gaseous Hg^0 was condensed to liquid mercury in an ice bath for collection (shown in Supplemental Fig. 24a). Approximately 87% mercury on the spent Se/PPS-I was recovered with this strategy. The selenium released during the decomposition of mercury was also condensed for recovery based on the variations in saturated vapour pressures (shown in Supplemental Fig. 24b).'

Schematic diagram of mercury recovery system

Photo of mercury and selenium recovered from spent Se/PPS-I. (a) liquid mercury, (b) selenium.

Comment 6:

The authors stated that the Se/PPS displayed ultralong-term real application potential. So, the mercury adsorption curve of Se/PPS should be provided, and the service lifetime when adopting in a fixed bed should be evaluated.

We thank the reviewer for this valuable suggestion. The Hg⁰ adsorption curve of Se/PPS has been added in the revised manuscript and supplementary materials (please refer to Fig. 7e and Supplemental Fig. 22). The same figures are shown below for your reference. According to the saturated Hg⁰ adsorption capacity and the adsorption breakthrough curve, the service time of Se/PPS when adopted in a typical coal-fired flue gas condition was estimated. Related discussions have been added to the manuscript (please refer to the 2nd paragraph in page 15). The same revised discussion is shown below for your reference.

‘A large Hg⁰ adsorption capacity was crucial for practical application of the monolithic Se/PPS-I in a fixed-bed since ultralong-term stability would prolong the service life and avoid frequent adsorbent replacement. As shown in Fig. 7e, almost no Hg⁰ was detected at the adsorbent bed outlet over 70 h with a specific gas flow rate of 1.13 m min⁻¹, even though the inlet Hg⁰ concentration was as high as 1000 μg m⁻³. The Hg⁰ adsorption capacity of Se/PPS-I was 1203.4 mg g⁻¹ when reaching 80% breakthrough (shown in Supplemental Fig. 22). Additionally, the adsorption rate was calculated as 1005.6 μg g⁻¹ min⁻¹. This Hg⁰ adsorption capacity and rate are the highest recorded values among various materials (shown in Fig. 7f and Supplemental Table S2)^{14, 19, 42, 43, 44, 45, 46, 47, 48, 49, 50, 51}. Based on the saturated Hg⁰ adsorption capacity and the adsorption breakthrough curve, the Hg⁰ concentration in coal-fired flue gas is expected to be reduced from approximately 20 μg m⁻³ to below 1.7 μg m⁻³ (i.e., the current most rigorous emission limit in the world) for 9940 h if Se/PPS-I was used with a real specific gas flow rate of 0.8 m min⁻¹.’

(a) Hg^0 adsorption curve, and (b) Hg^0 adsorption capacity of Se/PPS-I as a function of time

Comment 7:

Whether the selenium on Se/PPS would be easily to fall off by flue gas during long-term application, since the fall off of selenium not only affects the mercury removal performance but also causes selenium pollution?

We thank the reviewer for this insightful comment. We have tested the content of selenium lost from Se/PPS purging with N_2 for 14 days so as to confirm the stability of selenium during long-term application. The results have been added in the supplementary materials and denoted Supplemental Fig. 11 (the same figures are shown below for your reference). As shown, the weight change of Se/PPS after purging with N_2 for 14 days could be negligible. Meanwhile, we have determined the selenium content on the Se/PPS purging with N_2 for 14 days by inductively coupled plasma mass spectrometry. The selenium content on the purged Se/PPS was 2.81%, which was very close to that on the fresh Se/PPS (2.83%). This could fully demonstrate that selenium was firmly anchored on the PPS. Accordingly, the fall off of selenium during long-term application was insignificant in affecting the Hg^0 adsorption performance and causing selenium pollution. Related experimental methods and discussions have been added in the revised manuscript (1st paragraph on page 8) and the revised supplementary materials (2nd paragraph on page S3).

The weight change of Se/PPS purging with N₂ for 14 days

The selenium content of Se/PPS before and after purging with N₂ for 14 days (determined by inductively coupled plasma mass spectrometry)

Comment 8:

Whether the particulate matter in flue gas would cover the active sites (i.e., selenium), thus hindering the mercury removal performance of Se/PPS?

We thank the reviewer for this valuable comment. We have added experiments to investigate the effect of particulate matter on the Hg⁰ removal performance of Se/PPS. The Se/PPS was firstly covered by particulate matter (i.e., fly ash) collected from a coal-fired power plant, and then the Hg⁰ removal performance was tested. The result has been added to the revised supplementary materials and denoted

as Supplemental Fig. 21 (the same figure is shown below for your reference). As shown, very close Hg^0 removal performances of Se/PPS with and without covering by particulate matter were obtained. Thus, the inhibitive effect of particulate matter on the Hg^0 removal performance of Se/PPS can be negligible. Related discussions have also been added in the revised manuscript (please refer to the 1st paragraph in page 15). The same revised discussion is shown below for your reference.

‘The excellent performance of Se/PPS-I at 100–150 °C (Fig. 6e) implied that Se/PPS-I could serve as a dust filter bag material to achieve simultaneous removal of particulate matter and Hg^0 from flue gases, as illustrated in Supplemental Fig. 20. Supplemental Fig. 21 shows that the Hg^0 removal capacities of Se/PPS with and without coverage by particulate matter were similar, implying feasibility for long-term use under high-concentration dust conditions.’

(a) Se/PPS covered by fly ash, (b) Hg^0 removal performances of Se/PPS with and without covering by fly ash

Comment 9:

The authors adopted the pseudo-first-order and pseudo-second-order kinetic models to simulate the mercury adsorption behavior, but the kinetic models were not present in the manuscript.

We thank the reviewer for this great reminder. The kinetic models have been added in the revised supplementary materials (please see page S6 and page S7). Other typical kinetic models including

Intra-particle diffusion model and Elovich model have been also adopted to further investigate the Hg⁰ adsorption behaviors over Se/PPS. The same kinetic modes are shown below for your reference.

‘Description of Adsorption Kinetic Models

Pseudo-first-order model

This model is based on the mass balance. The Hg adsorption rate was proportional to the difference between the equilibrium capacity and the adsorbed amount at any time, as described following:

$$(1)$$

Equation (1) could be modified to the following equation based on the initial conditions of $t=0$ $q_t=0$ and $t=t$ $q_t=q_t$:

$$(2)$$

where q_t and q_e represent the adsorbed mercury amount at any time t and equilibrium time ($\mu\text{g}\cdot\text{g}^{-1}$). k_1 represents the rate constant (min^{-1}). q_e and k_1 can be attained by fitting the adsorption curve.

Pseudo-second-order model

The pseudo-second-order model represents that the surface diffusivity is inversely proportional to the square of concentration variation on sorbent surface, which could be described as following:

$$(3)$$

Equation (1) can be modified to the following equation based on the initial conditions of $t=0$ $q_t=0$ and $t=t$ $q_t=q_t$:

$$(4)$$

where k_2 represents the rate constant (min^{-1}). The term $k_2q_e^2$ was the initial adsorption rate.

Intra-particle diffusion model

The Intra-particle diffusion model assumes that the intra-particle diffusivity is constant and the diffusion direction is radial. The model can be interpreted by the following equation:

(6)

where k_{id} represents the intraparticle diffusion rate constant, C is proportional to the boundary layer.

Elovich model

The Elovich model assumes that sorption takes place in two phases: (1) a fast initial reaction associated with the movement of the sorbate to external sites, (2) a slower diffusion in and out of the microspores over sorbent. This model can be described by the following equation:

(7)

where α represents the initial rate, β is related to the extent of surface coverage and activation energy for chemisorption. If t is much larger than t_0 this equation is modified as follows:

(8)'

Reviewer #2

A brand-new method was developed in this work to achieve the efficient capture of Hg^0 from industrial flue gas. The most impressive novelty of this work is that it mimics the Hg^0 detoxification mechanism in human body, which has hardly been proposed in previous studies. The novelty of this work thus makes it suitable to be published on Nature Communications. I believe it can significantly extend the research horizons and raise wide research interests in related fields. The results are somehow interesting, but there are some notes that the authors have to pay attention to. The work is recommended to publish after a minor revision. Specific suggestions are as follows.

Comment 1:

Please provide experimental details. For instance, how to obtain the spent sample for XPS and TPD tests? The sorbent dosage, flue gas atmosphere and flue gas temperature should be also described clearly

for each set experiment, since the Hg^0 adsorption efficiency was dependent on these experimental parameters.

We thank the reviewer for this valuable suggestion. The sample pretreatment method for XPS and TPD tests has been added in page S4 of the revised supplementary materials. A table for summarizing the experimental conditions including flue gas atmosphere, flue gas temperature, gas hourly space velocity (GHSV) have been also presented in the revised supplementary materials and denoted as Table S1 (the same table is shown below for your reference).

Experimental conditions

Experiments	Sorbents	Gas components	Temperature (°C)	GHSV (m min ⁻¹)	Hg ⁰ concentration (μg m ⁻³)
Set I	PPS, Se/PPS-P, Se/PPS-I, SeO ₃ ²⁻ -PPS, Black Se powder, Red Se powder Se/PPS-I (0%, 0.1%,0.25%, 0.5%,1%NaOH concentration;	N ₂	125	5.66	1000
Set II	2.75%,5.5%,11%, 22%, 44% SeO ₃ ²⁻ concentration)	N ₂	125	5.66	1000
Set III	PPS, Se/PPS-I	N ₂	25,50, 75, 100, 125, 150	5.66	1000
Set IV	Se/PPS-I	N ₂	125	5.66	1000

80%N₂+20%O₂
 N₂+8%/14%/20%
 H₂O
 N₂+2000ppm/1%/
 2%SO₂
 N₂+5%O₂+8H₂O
 +2000ppmSO₂

Set V	Se/PPS-I (with and without covering by particulate matter)	N ₂	125	5.66	1000
Set VI	Se/PPS-I	N ₂	125	0.57, 2.83, 5.66, 11.32	1000
Set VII	Se/PPS-I	N ₂	125	1.13, 5.66	1000
Set VIII	Polyester Se/Polyester	N ₂	25, 50, 75, 100, 125, 150	5.66	1000

Comment 2:

DTA curves should be added to accompany with TG curve to better illustrate the thermal stability of the Se/PPS.

We thank the reviewer for this insightful suggestion. DTA curves have been added to accompany with TG curve in the revised manuscript and denoted as Fig. 4a and Supplemental Fig. 10. The same figures are shown below for your reference.

TG-DTA curve for (a) Se/PPS, (b) PPS, and (c) Se powder

Comment 3:

The Se/PPS was designed to be adopted in a fixed-bed reactor like filter bag type dust collector to achieve simultaneous removal of particulate matters and mercury from flue gas. A scheme for Hg^0 removal by Se/PPS was suggested to be provided.

We thank the reviewer for this insightful suggestion. A scheme for Hg^0 removal by Se/PPS has been added in the revised supplementary materials and denoted as Supplemental Fig. 20 (the same figure is shown below for your reference). Related discussions have been revised in the updated manuscript (please refer to the 1st paragraph in page 15). The same revision is shown below for your reference.

‘Additionally, as a monolithic adsorbent, Se/PPS-I was designed for use in a fixed-bed process. The excellent performance of Se/PPS-I at 100–150 °C (Fig. 6e) implied that Se/PPS-I could serve as a dust

filter bag material to achieve simultaneous removal of particulate matter and Hg^0 from flue gases, as illustrated in Supplemental Fig. 20.'

A proposed scheme for simultaneous removal of Hg^0 and dust by Se/PPS

Comment 4:

A comparative table should be provided to demonstrate the superior performance of Se/PPS compared to other sulfur/selenium functionalized substrates in removing Hg^0 , further proving the advanced nature of the biomimetic approach.

We thank the reviewer for this valuable suggestion. A comparison table showing the Hg^0 adsorption performances of Se/PPS and other adsorbents has been added in the supplementary materials and denoted Table S2 (the same table is shown below for your reference). Besides sulfur/selenium functionalized materials, other typical adsorbents including metal oxides, noble metals, minerals, etc., have been also included for a thorough comparison. As shown, the Hg^0 adsorption capacity of Se/PPS was the highest among all of benchmark materials. This table fully justifies the superior performance

of Se/PPS in removing Hg⁰. Related discussions have been revised in the updated manuscript (please refer to the 2nd paragraph in page 15).

Hg⁰ adsorption capacities of different adsorbents

Adsorbents	Carrier gas	Breakthrough threshold/Test time (h)	Hg ⁰ adsorption capacities (mg g ⁻¹)	Hg ⁰ concentration	References
Se/PPS	N ₂	80%/96h	1203.4	1000	This work
CuSe	N ₂	80%/13h	80.2	1000	1
CuSe/CAU-10-U	N ₂	70%/100h	302.23	100	2
Se/ZIF-8	N ₂	100%/NA	40.8	200	3
CuSe/ZIF-8	N ₂	100%/50h	315.2	200	3
WSe ₂	NA	NA/80h	30.6	560	4
CuSe/g-C ₃ N ₄	NA	80%/11h	9.1	180	5
Se/MIL-101	N ₂	80%/220h	148.19	1000	6
Fe ₃ O _{4-x} Se _y	SFG1	100%/50h	8.8	65	7
CuS	N ₂	80%/5h	103.73	1000	8
Fe/ZnS	Air	47%/60h	8.65	1600	9
Co _x Zn _{1-x} S	N ₂ +O ₂	50%/NA	46.01	1100	10
Co ₃ S ₄	N ₂	50%/24h	43.03	1300	11
Fly ash	NA	100%/3h	0.0103	10	12
S-AC	SFG2	49%/16.25 h	1.156	10	13
Cl-AC	Air	NA/8h	0.879	35	14
Br-AC	N ₂	100%/2.5h	1.53	NA	15

SBA-15-Ag	NA	NA	0.06	77.6	16
Ag-beads	N ₂	1%/NA	0.0002	447	17
Au-beads	N ₂	1%/NA	0.0029	447	17
Ag/Graphene	N ₂	NA	4.2	500	18
MnO ₂ @MOS	N ₂ +O ₂	20%/32h	1.52	300	19
CeO ₂ /MnO _x /TiO ₂	SFG3	NA	9.4	30-50	20
CeO ₂	N ₂	NA/2h	0.1	80	21
CeO ₂ /ZnIn ₂ S ₄	N ₂ +O ₂	NA/10h	0.9	80	21

SFG1: 4% O₂, 500 ppm SO₂ and 8% H₂O

SFG2: 14% CO₂, 6% O₂, 10% H₂O, 50 ppm HCl, 200 ppm SO₂ and 200 ppm NO

SFG3: 400 ppm NO and 400 ppm CO

Reference

1. Yang J, *et al.* Charge distribution modulation and morphology controlling of copper selenide for an enhanced elemental mercury adsorption activity in flue gas. *Chem. Eng. J.* **442**, 136145 (2022).
2. Jia T, Gu Y, Wu J, Li F. Copper selenide sensitized low-cost porous coordination polymers towards efficient capture trace gaseous elemental mercury. *Chem. Eng. J.* **457**, 141288 (2023).
3. Yang Z, *et al.* Nanosized copper selenide functionalized zeolitic imidazolate framework-8 (CuSe/ZIF-8) for efficient immobilization of gas-phase elemental mercury. *Adv. Funct. Mater.* **29**, 1807191 (2019).
4. Duan X, *et al.* Efficient immobilization and detoxification of gaseous elemental mercury by nanoflower/rod WSe₂/halloysite composite: Performance and mechanisms. *J. Hazard. Mater.* **458**, 131898 (2023).
5. Liu H, Ruan W, Zhang Z, Shen F, Zhou Y, Yang H. Dual 2-dimensional CuSe/g-C₃N₄ nano-heterostructure for boosting immobilization of elemental mercury in flue gas. *Chem. Eng. J.* **435**, 134696 (2022).
6. Yang J, *et al.* Selenium functionalized metal-organic framework MIL-101 for efficient and permanent sequestration of mercury. *Environ. Sci. Technol.* **53**, 2260-2268 (2019).
7. Yang Z, *et al.* Development of selenized magnetite (Fe₃O_{4-x}Se_y) as an efficient and recyclable trap for elemental mercury sequestration from coal combustion flue gas. *Chem. Eng. J.* **394**, 125022 (2020).

8. Meng F, *et al.* Cupric ion stabilized iron sulfide as an efficient trap with hydrophobicity for elemental mercury sequestration from flue gas. *Sep. Purif. Technol.* **330**, 125385 (2024).
9. Liao Y, Xia Y, Zou S, Liu P, Liang X, Yang S. In situ emergency disposal of liquid mercury leakage by Fe-containing sphalerite: Performance and reaction mechanism. *Ind. Eng. Chem. Res.* **56**, 153-160 (2017).
10. Liu W, Xu H, Liao Y, Wang Y, Yan N, Qu Z. Co-doped ZnS with large adsorption capacity for recovering Hg⁰ from non-ferrous metal smelting gas as a co-benefit of electrostatic demisters. *Environ. Sci. Pollu. R.* **27**, 20469-20477 (2020).
11. Quan Z, *et al.* Study on the regenerable sulfur-resistant sorbent for mercury removal from nonferrous metal smelting flue gas. *Fuel* **241**, 451-458 (2019).
12. Zhao Y, *et al.* Experimental study on fly ash capture mercury in flue gas. *Science China Technological Sciences* **53**, 976-983 (2010).
13. Hsi H, Tsai C, Lin K. Impact of surface functional groups, water vapor, and flue gas components on mercury adsorption and oxidation by sulfur-impregnated activated carbons. *Energy & Fuels* **28**, 3300-3309 (2014).
14. Zeng H, Jin F, Guo J. Removal of elemental mercury from coal combustion flue gas by chloride-impregnated activated carbon. *Fuel* **83**, 143-146 (2004).
15. Zhou Q, *et al.* Experimental and kinetic studies of gas-phase mercury adsorption by raw and bromine modified activated carbon. *Fuel. Process. Technol.* **134**, 325-332 (2015).
16. Xie Y, Yan B, Tian C, Liu Y, Liu Q, Zeng H. Efficient removal of elemental mercury (Hg⁰) by SBA-15-Ag adsorbents. *J. Mater. Chem. A* **2**, 17730-17734 (2014).
17. Luo G, *et al.* Carbon nanotube-silver composite for mercury capture and analysis. *Energy & Fuels* **24**, 419-426 (2010).
18. Xu H, *et al.* Regenerable Ag/graphene sorbent for elemental mercury capture at ambient temperature. *Colloids. Surfaces. A* **476**, 83-89 (2015).
19. Hao R, *et al.* Enhanced removal of elemental mercury using MnO₂-modified molecular sieve under microwave irradiation. *Chem. Eng. J.* **450**, 137997 (2022).
20. He J, Reddy G, Thiel S, Smirniotis P, Pinto N. Simultaneous removal of elemental mercury and NO from flue gas using CeO₂ modified MnO_x/TiO₂ materials. *Energy & Fuels* **27**, 4832-4839 (2013).
21. Jia T, *et al.* Nanosized ZnIn₂S₄ supported on facet-engineered CeO₂ nanorods for efficient gaseous elemental mercury immobilization. *J. Hazard. Mater.* **419**, 126436 (2021).

Comment 5:

At the current stage, the preparation parameters of the adsorbents are not clear. The author must specify the dosage of each precursor in the adsorbent preparation section to ensure the reproducibility of this work.

We thank the reviewer for this great reminder. We have clearly described the sample preparation parameters to ensure the reproducibility of this work (please refer to pages S2 and S3 of the supplementary materials). The same revision is shown below for your reference.

The *in-situ* synthesis procedure for Se/PPS was as follows. First, 2.6 mmol sodium selenite (Na_2SeO_3) was fully dissolved in deionized water and stirred for 0.5 h. Then, a piece of commercially purchased PPS fabrics (6 g) was immersed in the solution and oscillated for 6 h continuously. After that, 5.85 mmol glutathione (GSH, reduced form) was added into the solution and oscillated for another 6 h. Then, 12.5 mmol sodium hydroxide (NaOH) solution was dropwise added into the suspension to bring the pH above 12 and oscillated for 5 h at room temperature. Finally, the product was rinsed with deionized water before drying at 110 °C for 12 h. The as-prepared sample is denoted as Se/PPS-I. The dosage of selenium precursor and NaOH was also adjusted to synthesize Se/PPS-I samples with various Se coating mounts and morphologies. Polyester fabric without containing sulfur was also adopted to be functionalized by the *in-situ* synthesis method (denoted as Se/Polyester). In addition, the SeO_3^{2-} loaded PPS (denoted as SeO_3^{2-} /PPS) was prepared by the same procedure for obtaining Se/PPS-I but without adding GSH and NaOH.

As a reference, the post synthesis method was adopted to prepare Se/PPS-P. First, a suspension containing selenium particles was prepared by dissolving 2.6 mmol of Na_2SeO_3 , 5.85 mmol of GSH, and 12.5 mmol of NaOH in deionized water and stirring for 5 h. Then, a piece of PPS fabric (6 g) was immersed in the solution and oscillated for 5 h continuously. Finally, the product was rinsed with

deionized water several times and dried at 110 °C for 12 h. The dosages of selenium precursor and NaOH were the same as those for preparing Se/PPS-I samples.

The powdery selenium preparation procedure was same as that for Se/PPS but only without containing PPS. Specifically, 2.6 mmol Na_2SeO_3 was fully dissolved in deionized water and stirred for 0.5 h. After that, 5.85 mmol GSH (reduced form) was added into the solution and oscillated for another 6 h. Then, 12.5 mmol NaOH solution was dropwise added into the suspension to bring the pH above 12 and oscillated for 5 h at room temperature. Finally, the powdery selenium sample was obtained after centrifuging, rinsing with deionized water, and drying at 110 °C for 12 h.'

Comment 6:

The characterization of the adsorbent can be moved to the main content of the manuscript instead of placing it in the supplementary information. Additionally, specific conditions used to obtain characterization results can be further specified to support the reproducibility of this work.

We thank the reviewer for this valuable suggestion. We have moved the characterization results and related discussion to the main content of the manuscript (please refer to Figure 7 and Supplemental Fig. 19, the 2nd paragraph in page 14 and 1st paragraph in page 15). The same revision is shown below for your reference.

'Since Se/PPS-I was designed for use with different industrial flue gases, the Se/PPS-I should withstand harsh operating conditions. As shown in Fig. 7a, Se/PPS-I did not exhibit interference from typical flue gas components, including H_2O , SO_2 and particulate matter. The water contact angle above 130° indicated the hydrophobicity of Se/PPS-I (shown in Fig. 7b). As a result, the likelihood of H_2O covering the selenium surface was diminished, thus eliminating the adverse effect of H_2O on Hg^0 adsorption. Additionally, there were no obvious changes in the oxidation states of Se and S on the

fresh and pretreated Se/PPS (shown in Fig. 7c). Thus, unlike other adsorbents reported previously, such as activated carbons, metal oxides, and noble metals, which were partially deactivated by the presence of H₂O and SO₂,^[31] Se/PPS-I maintained stable Hg⁰ adsorption in flue gases containing 20% H₂O and 2% SO₂ (shown in Supplemental Fig. 19).’

Specific conditions used to obtain characterization results have been described clearly to ensure the reproducibility of this work (please refer to the 2nd paragraphs in page S3 and 1st paragraphs in page S4 in the supplementary materials). The same revision is shown below for your reference.

‘The spent sample was characterized by XPS to investigate the surface chemistry variation. The fresh sample was pretreated by 1 L·min⁻¹ N₂ containing 1000 µg·m⁻³ Hg⁰ at 125 °C for 2 or 50 h and purging by 1 L·min⁻¹ N₂ to remove the unstable mercury to obtain a mercury-laden sample. The mercury adsorption product on the spent sorbent was studied by temperature programmed desorption (TPD) experiments, where the mercury species could be identified by comparing the desorption characteristics with that of the reference pure mercury compounds. The sample adopting for the TPD experiment was same with that for XPS analysis.’

Comment 7:

In this work, the characterization of adsorbents for Hg⁰ removal performance seems to be primarily based on the Hg⁰ removal efficiency and Hg⁰ adsorption capacity presented in this study. However, the authors only provide the calculation method for Hg⁰ adsorption capacity. Please add the calculation method for Hg⁰ removal efficiency.

We thank the reviewer for this great reminder. The Hg⁰ removal efficiency has been defined in the revised supplementary materials (please refer to the 1st paragraph in page S5). The same equation to calculate the Hg⁰ removal efficiency is shown below for your reference.

'The Hg⁰ adsorption capacity and Hg⁰ removal efficiency were calculated by the following equations.

$$Q = \frac{1}{A} \int_{t_1}^{t_2} (C_{in} - C_{out}) \times f \times dt \quad (1)$$

$$\eta = \frac{\int_0^t C_{in}^0 - \int_0^t C_{out}^0}{\int_0^t C_{in}^0} \quad (2)$$

where Q is the Hg⁰ adsorption capacity (mg Hg • g⁻¹ adsorbent), η is the Hg⁰ removal efficiency (%), A is the volume of sorbent (m³), f is the gas flow rate (m³ h⁻¹), t is the accumulated time of every set experiment (h) and $t=120$ min when calculating the adsorption efficiency, C_{in} and C_{out} is the inlet and outlet Hg⁰ concentration (µg m⁻³).

Comment 8:

Due to the ability of sulfur in PPS to adsorb Hg⁰, it is unclear whether there is still mercury sulfide present after adsorption saturation. I suggest that the authors further investigate the mercury species on Se/PPS with depleted selenium sites through long-term mercury pretreatment to firmly establish the role of sulfur in facilitating the immobilization of mercury on selenium through transport proteins.

We thank the reviewer for this insightful comment. We have conducted two sets of additional experiments to clarify the role of sulphur in facilitating the immobilization of mercury.

First, we compared the Hg⁰ removal performances of polyester, PPS, Se/polyester, and Se/PPS. The two supporters (polyester and PPS) as adopted shared similar tangled structures, while polyester contains no sulphur ligand. The results have been added in the supplementary materials and denoted Supplemental Fig. 17 (the same figures are shown below for your reference). As shown, when raising the reaction temperature, the Hg⁰ removal performances of PPS improved, but the Hg⁰ removal performance of polyester was slightly compromised. This phenomenon indicates that Hg⁰ adsorption

in PPS was primarily attributed to chemical interaction in the presence of sulfur ligands, because chemical interaction can be improved by higher reaction temperature with more energy input. However, for polyester, the physisorption effects might dominate, and higher temperature caused the desorption of physisorbed products. The observations indicate that, in the absence of Se, Hg^0 might chemically interact with sulfur ligands in PPS at relatively high temperature. Such chemical interaction was further supported by another controlled set of experiment between Se/polyester and Se/PPS. As shown in the following figure, Se/polyester and Se/PPS exhibited comparable Hg^0 removal performances at temperatures lower than 75 °C, which suggests that, before chemical interaction between mercury and sulphur was adequately activated, Se/polyester and Se/PPS did not vary in Hg^0 removal capacity. Contrarily, when the reaction temperature raised to higher than 75 °C, a temperature coincided with the activation temperature of chemical interaction, Se/PPS significantly outperformed Se/polyester for Hg^0 capture. The above-mentioned observations are performance proof of the facilitating role of sulphur ligands in promoting the Hg^0 removal on Se/PPS.

Hg⁰ adsorption performances of different samples at varied temperatures. (a) PPS, (b) Se/PPS-I, (c) Polyester, (d) Se/Polyester

Second, we have conducted additional mercury temperature programmed desorption/decomposition (Hg-TPD) experiments to confirm the migration of transient mercury-sulfur species to selenium sites for permanent immobilization. The results have been added in the revised manuscript and denoted Fig. 8(a) (the same figures are shown below for your reference). Four different Hg-laden samples were prepared. Specifically, the pristine PPS and Se/PPS-I were pretreated by 1 L·min⁻¹ N₂ containing 1000 μg·m⁻³ Hg⁰ at 125 °C for 2 or 50 h and purging by 1 L·min⁻¹ N₂ to remove the unstable mercury to obtain a mercury-laden sample. Besides, Hg-laden selenium powder mechanically mixed with PPS (i.e., selenium powder+PPS) was also used as a reference. Among the four samples, the Hg-laden selenium powder + PPS exhibited two characteristic decomposition peaks located at ~ 160 °C and 230 °C. By comparing the Hg-TPD patterns of pristine PPS, unsaturated Se/PPS-I (adsorption of Hg⁰ for

2 h), and selenium powder + PPS, it was found that the characteristic peak at ~ 160 °C was attributed to the chemical interaction between PPS-sulphur and mercury (no physisorption was accounted considering the high pretreatment and decomposition temperature), and the one centered at ~ 230 °C was ascribed to mercury-selenium interaction (i.e., HgSe) because this peak did not occur in the TPD pattern of Hg-laden PPS. It was found that no characteristic peak accounting for mercury-PPS interaction was observed in unsaturated Se/PPS-I because, when the sample was far from being saturated, and the amount of active selenium was abundant in the sample, mercury interacted with PPS supporter, if any, would be immediately transferred to mercury selenide on the surface of Se/PPS-I. However, if we increased the pretreatment time and consumed most of the active selenium sites in Se/PPS-I, the interaction between PPS and mercury could be observed, which was manifested by the occurrence of 160 °C peak in the saturated Se/PPS-I (adsorption of Hg^0 for more than 50 h). This characteristic proof further supports that the PPS-sulphur did interact with mercury, and the absence of mercury-sulfur intermediates in unsaturated Se/PPS is mainly ascribed to the spontaneous and rapid transformation of the intermediates into the mercury selenide final product when selenium sites were abundant.

TPD patterns for spent PPS, Se powder plus PPS, Se/PPS (unsaturation), and Se/PPS (saturation)

The above two sets of experiments could establish the role of sulfur in facilitating the

immobilization of mercury on selenium through transport carrier, i.e., Hg^0 was firstly weakly-adsorbed on sulfur and then captured by selenium for permanent immobilization as HgSe . However, since the mercury species pre-adsorbed on the sulfur in PPS would be immediately consumed, directly detecting the transition road between mercury-sulfur and mercury-selenium is extremely difficult with current techniques. Related discussions have been revised in the updated manuscript (please refer to the 2nd and in page 16, 1st and 2nd paragraphs in page 17, 1st paragraph in page 18).

Comment 9:

The conclusion section mainly provides some general comments on this work. Please provide more quantitative descriptions of the main findings of this study in the conclusion section.

We thank the reviewer for this valuable suggestion. We have rewritten the conclusion section to provide more quantitative descriptions of the main findings of this study (please refer to the 2nd paragraph in page 19). The same revision is shown below for your reference.

‘In conclusion, this work demonstrated for the first time the feasibility of enhancing Hg^0 adsorption on functionalized substrates via a biomimetic pathway. The population density, distribution, and crystalline form of selenium were rationally regulated by adjusting the concentrations of NaOH and SeO_3^{2-} from 0% to 1% and 2.75% to 44%, respectively. At a reaction temperature higher than 100 °C, which was within the operation temperature range for fabric fibers under practical flue gas cleaning scenarios, the resultant Se/PPS-I displayed a Hg^0 adsorption capacity and uptake rate of 1621.9 mg g^{-1} and 1005.6 $\mu\text{g g}^{-1} \text{min}^{-1}$, respectively. The excellent Hg^0 adsorption performance of Se/PPS-I was attributed to the plentiful sulfur sites in PPS that served as buffers for Hg^0 transport to the adjacent selenium. The resistance of Se/PPS-I to flue gas interference enables ultralong-term use under harsh flue gas conditions, and it is expected to serve for approximately 10000 h without changing the

adsorbent. This work developed an effective Hg^0 adsorbent and provided guidance for biomimetic design of advanced functional filters for pollutant abatement.’

Reviewer #3

In this work, the authors proposed a new strategy for mercury decontamination from industrial flue gas, a serious task for public health and environmental societies. The results and the mechanism in this study are reasonable. The manuscript can be accepted after revision. Some comments are listed as follows:

Comment 1:

Please avoid the use of exaggerated words like “significantly”, “excellent”, “obvious”, etc. because the Nature Communications is a journal valuing scientific expressions.

We thank the reviewer for this great reminder. We have refrained from using subjective language when referring to the scientific findings, and the related findings were described by quantitative conclusion in the revised manuscript.

Comment 2:

The Abstract mentioned that the Hg^0 adsorption capacity and adsorption rate of the Se/PPS are record-breaking high values. More data from previous literatures should be included to thoroughly demonstrate the superiority of the resultant Se/PPS. Moreover, the test conditions including sorbent dosage, Hg^0 concentration and test time should be presented when comparing the Hg^0 adsorption capacity and adsorption rate.

We thank the reviewer for this valuable suggestion. More data from previous literatures have been

added in the revised manuscript to thoroughly demonstrate the superiority of Hg⁰ adsorption performances of Se/PPS. Different kinds of adsorbents including sulfur/selenium functionalized materials, metal oxides, noble metals, minerals, etc., have been included for a thorough comparison. A comparison table showing the Hg⁰ adsorption performances of Se/PPS and other adsorbents has been added in the supplementary materials and denoted Table S2 (the same table is shown below for your reference). The test conditions including carrier gas, Hg⁰ concentration and test time have been presented in the table when comparing the Hg⁰ adsorption capacity and adsorption rate. As shown, the Hg⁰ adsorption capacity of Se/PPS was the highest among all of benchmark materials. This table fully justifies the superior performance of Se/PPS in removing Hg⁰. Related discussions have been revised in the updated manuscript (please refer to the 2nd paragraph in page 15).

Hg⁰ adsorption capacities of different adsorbents

Adsorbents	Carrier gas	Breakthrough threshold/Test time (h)	Hg ⁰ adsorption capacities (mg g ⁻¹)	Hg ⁰ concentration	References
Se/PPS	N ₂	80%/96h	1203.4	1000	This work
CuSe	N ₂	80%/13h	80.2	1000	1
CuSe/CAU-10-U	N ₂	70%/100h	302.23	100	2
Se/ZIF-8	N ₂	100%/NA	40.8	200	3
CuSe/ZIF-8	N ₂	100%/50h	315.2	200	3
WSe ₂	NA	NA/80h	30.6	560	4
CuSe/g-C ₃ N ₄	NA	80%/11h	9.1	180	5
Se/MIL-101	N ₂	80%/220h	148.19	1000	6
Fe ₃ O _{4-x} Se _y	SFG1	100%/50h	8.8	65	7

CuS	N ₂	80%/5h	103.73	1000	8
Fe/ZnS	Air	47%/60h	8.65	1600	9
Co _x Zn _{1-x} S	N ₂ +O ₂	50%/NA	46.01	1100	10
Co ₃ S ₄	N ₂	50%/24h	43.03	1300	11
Fly ash	NA	100%/3h	0.0103	10	12
S-AC	SFG2	49%/16.25 h	1.156	10	13
Cl-AC	Air	NA/8h	0.879	35	14
Br-AC	N ₂	100%/2.5h	1.53	NA	15
SBA-15-Ag	NA	NA	0.06	77.6	16
Ag-beads	N ₂	1%/NA	0.0002	447	17
Au-beads	N ₂	1%/NA	0.0029	447	17
Ag/Graphene	N ₂	NA	4.2	500	18
MnO ₂ @MOS	N ₂ +O ₂	20%/32h	1.52	300	19
CeO ₂ /MnO _x /TiO ₂	SFG3	NA	9.4	30-50	20
CeO ₂	N ₂	NA/2h	0.1	80	21
CeO ₂ /ZnIn ₂ S ₄	N ₂ +O ₂	NA/10h	0.9	80	21

SFG1: 4% O₂, 500 ppm SO₂ and 8% H₂O

SFG2: 14% CO₂, 6% O₂, 10% H₂O, 50 ppm HCl, 200 ppm SO₂ and 200 ppm NO

SFG3: 400 ppm NO and 400 ppm CO

Reference

1. Yang J, *et al.* Charge distribution modulation and morphology controlling of copper selenide for an enhanced elemental mercury adsorption activity in flue gas. *Chem. Eng. J.* **442**, 136145 (2022).
2. Jia T, Gu Y, Wu J, Li F. Copper selenide sensitized low-cost porous coordination polymers towards efficient capture trace gaseous elemental mercury. *Chem. Eng. J.* **457**, 141288 (2023).

3. Yang Z, *et al.* Nanosized copper selenide functionalized zeolitic imidazolate framework-8 (CuSe/ZIF-8) for efficient immobilization of gas-phase elemental mercury. *Adv. Funct. Mater.* **29**, 1807191 (2019).
4. Duan X, *et al.* Efficient immobilization and detoxification of gaseous elemental mercury by nanoflower/rod WSe₂/halloysite composite: Performance and mechanisms. *J. Hazard. Mater.* **458**, 131898 (2023).
5. Liu H, Ruan W, Zhang Z, Shen F, Zhou Y, Yang H. Dual 2-dimensional CuSe/g-C₃N₄ nano-heterostructure for boosting immobilization of elemental mercury in flue gas. *Chem. Eng. J.* **435**, 134696 (2022).
6. Yang J, *et al.* Selenium functionalized metal-organic framework MIL-101 for efficient and permanent sequestration of mercury. *Environ. Sci. Technol.* **53**, 2260-2268 (2019).
7. Yang Z, *et al.* Development of selenized magnetite (Fe₃O_{4-x}Se_y) as an efficient and recyclable trap for elemental mercury sequestration from coal combustion flue gas. *Chem. Eng. J.* **394**, 125022 (2020).
8. Meng F, *et al.* Cupric ion stabilized iron sulfide as an efficient trap with hydrophobicity for elemental mercury sequestration from flue gas. *Sep. Purif. Technol.* **330**, 125385 (2024).
9. Liao Y, Xia Y, Zou S, Liu P, Liang X, Yang S. In situ emergency disposal of liquid mercury leakage by Fe-containing sphalerite: Performance and reaction mechanism. *Ind. Eng. Chem. Res.* **56**, 153-160 (2017).
10. Liu W, Xu H, Liao Y, Wang Y, Yan N, Qu Z. Co-doped ZnS with large adsorption capacity for recovering Hg⁰ from non-ferrous metal smelting gas as a co-benefit of electrostatic demisters. *Environ. Sci. Pollu. R.* **27**, 20469-20477 (2020).
11. Quan Z, *et al.* Study on the regenerable sulfur-resistant sorbent for mercury removal from nonferrous metal smelting flue gas. *Fuel* **241**, 451-458 (2019).
12. Zhao Y, *et al.* Experimental study on fly ash capture mercury in flue gas. *Science China Technological Sciences* **53**, 976-983 (2010).
13. Hsi H, Tsai C, Lin K. Impact of surface functional groups, water vapor, and flue gas components on mercury adsorption and oxidation by sulfur-impregnated activated carbons. *Energy & Fuels* **28**, 3300-3309 (2014).
14. Zeng H, Jin F, Guo J. Removal of elemental mercury from coal combustion flue gas by chloride-impregnated activated carbon. *Fuel* **83**, 143-146 (2004).
15. Zhou Q, *et al.* Experimental and kinetic studies of gas-phase mercury adsorption by raw and bromine modified activated carbon. *Fuel. Process. Technol.* **134**, 325-332 (2015).
16. Xie Y, Yan B, Tian C, Liu Y, Liu Q, Zeng H. Efficient removal of elemental mercury (Hg⁰) by SBA-15-Ag adsorbents. *J. Mater. Chem. A* **2**, 17730-17734 (2014).
17. Luo G, *et al.* Carbon nanotube-silver composite for mercury capture and analysis. *Energy & Fuels* **24**, 419-426 (2010).

18. Xu H, *et al.* Regenerable Ag/graphene sorbent for elemental mercury capture at ambient temperature. *Colloids. Surfaces. A* **476**, 83-89 (2015).
19. Hao R, *et al.* Enhanced removal of elemental mercury using MnO₂-modified molecular sieve under microwave irradiation. *Chem. Eng. J.* **450**, 137997 (2022).
20. He J, Reddy G, Thiel S, Smirniotis P, Pinto N. Simultaneous removal of elemental mercury and NO from flue gas using CeO₂ modified MnO_x/TiO₂ materials. *Energy & Fuels* **27**, 4832-4839 (2013).
21. Jia T, *et al.* Nanosized ZnIn₂S₄ supported on facet-engineered CeO₂ nanorods for efficient gaseous elemental mercury immobilization. *J. Hazard. Mater.* **419**, 126436 (2021).

Comment 3:

The powdery selenium was adopted as a reference to compare the Hg⁰ adsorption efficiency with Se/PPS. Whether the powdery selenium a commercial sample? If so, the powdery selenium prepared with the same method adopted in this work should be prepared and the Hg⁰ adsorption efficiency of it should be tested.

We thank the reviewer for this insightful comment. The powdery selenium adopted for comparing the Hg⁰ adsorption efficiency with Se/PPS was prepared in our laboratory but not a commercial sample. We have clearly described the powdery selenium preparation method in the supplementary materials (please refer to the 3rd paragraph in page S2). The same revision is shown below for your reference.

‘The powdery selenium preparation procedure was same as that for Se/PPS but only without containing PPS. Specifically, 2.6 mmol Na₂SeO₃ was fully dissolved in deionized water and stirred for 0.5 h. After that, 5.85 mmol GSH (reduced form) was added into the solution and oscillated for another 6 h. Then, 12.5 mmol NaOH solution was dropwise added into the suspension to bring the pH above 12 and oscillated for 5 h at room temperature. Finally, the powdery selenium sample was obtained after centrifuging, rinsing with deionized water, and drying at 110 °C for 12 h.’

Comment 4:

What is the dosage of powdery selenium adopted for Hg^0 adsorption test (shown in Figure 6a), and what is the level compared with the amount of selenium on Se/PPS? It should be noteworthy that the samples were in different forms (i.e., powder and monolith). Thus, same amount of selenium should be adopted when comparing the Hg^0 adsorption performance.

We thank the reviewer for this great comment. We have normalized the selenium dosage when conducting the Hg^0 adsorption tests by powdery selenium and Se/PPS. Specifically, 108 mg Se/PPS with a selenium content of 2.83% was adopted for the Hg^0 adsorption test, i.e., 3.1 mg selenium existed on the Se/PPS. Same amount of powdery selenium (i.e., 3.1 mg) was adopted when conducting the Hg^0 adsorption test. We have clearly described the experimental details in the revised supplementary materials (please refer to the 1st paragraph in page S5).

Comment 5:

The Se/PPS displayed a mercury recovery feature owing to its great Hg^0 enrichment capacity. Mercury was recovered by decomposing the mercury adsorption products (i.e., HgSe) at high temperature. What is the decomposition temperature for mercury recovery. The volatile selenium would be released accompanying with the decomposition of HgSe at high temperature demonstrating by the TG results. Thus, how to dispose the released selenium since it is also an atmosphere pollutant.

We thank the reviewer for this constructive comment. The decomposition temperature for mercury recovery was set as 300 °C according to the TPD result, i.e., mercury on the spent Se/PPS-I was decomposed to gaseous Hg^0 in the temperature range of 150-300 °C. As commented, the selenium on Se/PPS would be released accompanied with the decomposition of HgSe . Thus, both mercury and

selenium were condensed for recovery based on their different saturated vapor pressure, as schematically illustrated by the following figure. It is observed that mercury and selenium can be deposited in the condensers and collected. The above figures have been added in the supplementary materials (please refer to Supplemental Fig. 23 and Fig. 24). Related experimental methods and discussions have been added in the revised manuscript (please refer to the 2nd paragraph in page 15 and 1st paragraph in page S6 in the supplementary materials). The same revision is shown below for your reference.

‘The enriched mercury on Se/PPS-I can be recovered as liquid mercury metal with a device capable of rapidly decomposing mercury compounds and efficiently separating liquid mercury (shown in Supplemental Fig. 23). Specifically, the decomposition temperature was set as 300 °C based on the TPD results, i.e., the mercury on the spent Se/PPS-I was decomposed to gaseous Hg⁰ in the temperature range 150–300 °C. The gaseous Hg⁰ was condensed to liquid mercury in an ice bath for collection (shown in Supplemental Fig. 24a). Approximately 87% mercury on the spent Se/PPS-I was recovered with this strategy. The selenium released during the decomposition of mercury was also condensed for recovery based on the variations in saturated vapour pressures (shown in Supplemental Fig. 24b).’

‘Mercury and selenium recovery

The recovery of mercury from spent Se/PPS was conducted using a device capable of rapidly decomposing mercury adsorption products to gaseous Hg⁰, which was then condensed to liquid mercury for collection (as illustrated by Supplemental Fig. 23). The decomposition temperature was determined by a TPD experiment, i.e., the temperature which could guarantee the full decomposition of mercury from spent Se/PPS. The spent Se/PPS was placed at a furnace to decompose the mercury adsorption products (i.e., HgSe), and the gaseous Hg⁰ was condensed to liquid mercury in an ice-bath.

Meanwhile, the selenium on Se/PPS would be released accompanying with the decomposition of HgSe. Thus, both mercury and selenium were condensed for recovery based on their different saturated vapor pressure.'

Schematic diagram of mercury and selenium recovery system

Photo of mercury and selenium recovered from spent Se/PPS-I. (a) liquid mercury, (b) selenium

Comment 6:

The stable Hg⁰ adsorption performances under harsh conditions were crucial for real-world applications. The authors stated that the Se/PPS displayed excellent anti-interference ability to general detrimental flue gas components (e.g., SO₂ and H₂O). The detrimental role of these flue gas components in Hg⁰ adsorption might be indeed inapparent during a short experiment time. However, the material was designed to be adopted in a fix-bed reactor, the service time would be generally

several months. Thus, I commend the authors could conduct the Hg^0 adsorption test with high SO_2 and H_2O concentration to further solid the above statement.

We thank the reviewer for this insightful comment and suggestion. The effects of SO_2 and H_2O on the Hg^0 adsorption performance of Se/PPS have been reconfirmed through conducting the Hg^0 adsorption test under high SO_2 and H_2O concentration, the results of which have been added in the revised manuscript and denoted Fig. 7a and Supplemental Fig. 19 (the same figure is shown below for your reference). Related experimental details and discussions have been added in the revised manuscript (the 2nd paragraph in page 14 and the 1st paragraph in page 15) and the supplementary materials (Table S1 and the 1st paragraph in page S5). The same revision is shown below for your reference.

‘Since Se/PPS-I was designed for use with different industrial flue gases, the Se/PPS-I should withstand harsh operating conditions. As shown in Fig. 7a, Se/PPS-I did not exhibit interference from typical flue gas components, including H_2O , SO_2 and particulate matter. The water contact angle above 130° indicated the hydrophobicity of Se/PPS-I (shown in Fig. 7b). As a result, the likelihood of H_2O covering the selenium surface was diminished, thus eliminating the adverse effect of H_2O on Hg^0 adsorption. Additionally, there were no obvious changes in the oxidation states of Se and S on the fresh and pretreated Se/PPS (shown in Fig. 7c). Thus, unlike other adsorbents reported previously, such as activated carbons, metal oxides, and noble metals, which were partially deactivated by the presence of H_2O and SO_2 ,^[31] Se/PPS-I maintained stable Hg^0 adsorption in flue gases containing 20% H_2O and 2% SO_2 (shown in Supplemental Fig. 19).’

Hg⁰ adsorption performance of Se/PPS under different atmospheres. (a) Hg⁰ adsorption efficiency of Se/PPS under different flue gas components, effect of (b) different concentration H₂O, (c) different concentration SO₂, and (d) co-existing H₂O and SO₂ on the Hg⁰ adsorption performances of Se/PPS

Comment 7:

The pseudo-first and pseudo-second kinetic models were adopted to simulate the Hg⁰ adsorption behaviors over Se/PPS. Other kinetic models like intra-particle diffusion model and elovich model were recommended to be adopted to further investigate the Hg⁰ adsorption process.

We thank the reviewer for this constructive comment. The Intra-particle diffusion model and Elovich model have been adopted to further investigate the Hg⁰ adsorption process. The models are

shown as follows. The results have been added in the supplementary materials and denoted Supplementary Fig. 18 (the same figures are shown below for your reference). Among various typical kinetic models, the pseudo-first model and intra-particle diffusion model were closest to simulate the Hg^0 adsorption process over Se/PPS-I. Related model descriptions and discussions have been added in the revised manuscript (the 3rd paragraph in page 13 and the 1st paragraph in page 14) and the supplementary materials (pages S6-S7). The same revisions are shown below for your reference.

‘Description of sorption kinetic models

Pseudo-first-order model

This model is based on the mass balance. The Hg adsorption rate was proportional to the difference between the equilibrium capacity and the adsorbed amount at any time, as described following:

$$(1)$$

Equation (1) could be modified to the following equation based on the initial conditions of $t=0$ $q_t=0$ and $t=t$ $q_t=q_t$:

$$(2)$$

where q_t and q_e represent the adsorbed mercury amount at any time t and equilibrium time ($\mu\text{g}\cdot\text{g}^{-1}$). k_1 represents the rate constant (min^{-1}). q_e and k_1 can be attained by fitting the adsorption curve.

Pseudo-second-order model

The pseudo-second-order model represents that the surface diffusivity is inversely proportional to the square of concentration variation on sorbent surface, which could be described as following:

$$(3)$$

Equation (1) can be modified to the following equation based on the initial conditions of $t=0$ $q_t=0$ and $t=t$ $q_t=q_t$:

$$(4)$$

where k_2 represents the rate constant (min^{-1}). The term $k_2q_e^2$ was the initial adsorption rate.

Intra-particle diffusion model

The intra-particle diffusion model assumes that the intra-particle diffusivity is constant and the diffusion direction is radial. The model can be interpreted by the following equation:

$$(6)$$

where k_{id} represents the intraparticle diffusion rate constant, C is proportional to the boundary layer.

Elovich model

The Elovich model assumes that sorption takes place in two phases: (1) a fast initial reaction associated with the movement of the sorbate to external sites, (2) a slower diffusion in and out of the microspores over sorbent. This model can be described by the following equation:

$$(7)$$

where α represents the initial rate, β is related to the extent of surface coverage and activation energy for chemisorption. If t is much larger than t_0 this equation is modified as follows:

$$(8)'$$

‘The Hg^0 adsorption on Se/PPS-I were fitted with typical kinetic models, the results of which are shown in Supplemental Fig. 18. The pseudo-first-order and intra-particle diffusion models fitted well with the experimental data, with correlation coefficient (R^2) of 0.97 and 0.99, respectively. This suggests that Hg^0 adsorption on Se/PPS-I was controlled by external and internal mass transfer.’

Kinetics models for Hg⁰ adsorption over Se/PPS-I. (a) Pseudo-first-order, (b) Pseudo-second-order kinetic models, (c) Intra-particle diffusion model, and (d) Elovich model

Comment 8:

The mercury decomposition characteristic of Hg-laden PPS investigated by TPD experiment was described in lines Hg-TPD product of Hg desorption on PPS has been described earlier in lines 212-217, but the results were described again in the subsequent reaction mechanism analyses. Thus, it is suggested to reorganize the relevant discussion.

We thank the reviewer for this constructive suggestion. We have reorganized the relevant discussion regarding the TPD experiments (please refer to the 2nd paragraph in page 17). The same revision is shown below for your reference.

‘TPD experiments were conducted to confirm the migration of transient mercury-sulfur species to

selenium sites for permanent immobilization. As shown in Fig. 8a, the Hg-laden selenium powder + PPS (i.e., selenium powder mechanically mixed with PPS) exhibited two characteristic decomposition peaks located at ~ 160 °C and 230 °C. By comparing the Hg-TPD patterns of pristine PPS, unsaturated Se/PPS-I (adsorption of Hg⁰ for 2 h), and selenium powder + PPS, it was found that the characteristic peak at ~ 160 °C was attributed to the chemical interaction between PPS-sulphur and mercury (no physisorption was accounted considering the high pretreatment and decomposition temperature), and the one centered at ~ 230 °C was ascribed to HgSe¹². It was found that no characteristic peak accounting for mercury-PPS interaction was observed in unsaturated Se/PPS-I because, when the sample was far from being saturated, and the amount of active selenium was abundant in the sample, mercury interacted with PPS supporter, if any, would be immediately transferred to mercury selenide on the surface of Se/PPS-I. However, if increased the pretreatment time and consumed most of the active selenium sites in Se/PPS-I, the interaction between PPS and mercury could be observed, which was manifested by the occurrence of 160 °C peak in the saturated Se/PPS-I (adsorption of Hg⁰ for more than 50 h). This further supports that the PPS-sulphur did interact with mercury, and the absence of mercury-sulfur intermediates in unsaturated Se/PPS is mainly ascribed to the spontaneous and rapid transformation of the intermediates into the mercury selenide final product when selenium sites were abundant. Thus, it was speculated that the plentiful sulfur in Se/PPS-I provided bridges to intercept Hg⁰, and the weakly adsorbed mercury subsequently migrated to the selenium interfaces for permanent immobilization. ’

REVIEWERS' COMMENTS

Reviewer #1 (Remarks to the Author):

The present manuscript on mercury immobilization by Se functionalized PPS is highly effective with the highest recorded values of Hg⁰ adsorption capacity as well as uptake rate (1621.9 mg·g⁻¹ and 1005.6 ug·g⁻¹·min⁻¹), exhibiting a distinctive advantage than other materials. The paper is well written and very clear. The presentation and analysis of results are interesting with good comparison and well structured. I recommend accepting the manuscript.

Reviewer #2 (Remarks to the Author):

This article proposed a brand-new method to achieve the efficient capture of Hg⁰ from industrial flue gas. Unlike previous studies, it presents a novel detoxification mechanism of mercury in the human body. The nine questions raised during the review process have all been explained by the author with detailly and reasonably. Supplementary materials have been added, and modifications have been made accordingly in the article. The work is recommended to publish on Nature Communications after above revisions.

Reviewer #3 (Remarks to the Author):

The manuscript can be accepted in this state.